# Detection of critical $PM_{2.5}$ emission sources and their contributions to a heavy haze episode in Beijing, China, using an adjoint model

Shixian Zhai[1], Xingqin An[2], Tianliang Zhao[1], Zhaobin Sun[3,4], Qing Hou[2], Chao Wang[2]

[1]Key Laboratory for Aerosol-Cloud-Precipitation of China Meteorological Administration, Collaborative Innovation Center on Forecast and Evaluation of Meteorological Disasters, School of Atmospheric Physics, Nanjing University of Information Science & Technology, Nanjing 210044, China

[2]State Key Laboratory of Severe Weather, Key Laboratory of Atmospheric Chemistry of CMA, Chinese Academy of Meteorological Sciences, Beijing 100081, China

[3]Institute of Urban Meteorology, China Meteorological Administration, Beijing 100089, China

[4]Environmental Meteorology Forecast Center of Beijing-Tianjin-Hebei, China Meteorological Administration, Beijing, 100089, China

*Correspondence to*: Xingqin An (anxq@camscma.cn) and Tianliang Zhao (tlzhao@nuist.edu.cn)

**Abstract.** Air pollution sources and their regional transport are important issues for air quality control. The Global–Regional Assimilation and Prediction System coupled with the China Meteorological Administration Unified Atmospheric Chemistry Environment (GRAPES–CUACE) aerosol adjoint model was applied to detect the sensitive primary emission sources of a haze episode in Beijing occurring between 19 and 21 November 2012. The high $PM_{2.5}$ concentration peaks occurring at 05:00 and 23:00 LT (GMT+8) over Beijing Municipality on November 21, 2012, were set as the cost functions for the aerosol adjoint model. The critical emission regions of the first $PM_{2.5}$ concentration peak were tracked to the west and south of Beijing, with 2 to 3 d cumulative transport of air pollutants to Beijing. The critical emission regions of the second peak were mainly located to the south of Beijing, where southeasterly moist air transport led to the hygroscopic growth of particles and pollutant convergence in front of the Taihang Mountains during the daytime on November 21. The temporal variations in the sensitivity coefficients for the two $PM_{2.5}$ concentration peaks revealed that the response time of the onset of Beijing haze pollution from the local primary emissions is approximately 1–2 h and that from the surrounding primary emissions it is approximately 7–12 h. The upstream Hebei Province has the largest impact on the two $PM_{2.5}$ concentration peaks, and the contribution of emissions from Hebei Province to the first $PM_{2.5}$ concentration peak (43.6%) is greater than that to the second $PM_{2.5}$ concentration peak (41.5%). The second largest influential province for the 05:00 LT $PM_{2.5}$ concentration peak is Beijing (31.2%), followed by Shanxi (9.8%), Tianjin (9.8%), and Shandong (5.7%). The second largest influential province for the 23:00 LT $PM_{2.5}$ concentration peak is Beijing (35.7%), followed by Shanxi (8.1%), Shandong (8.0%), and Tianjin (6.7%). The adjoint model results were compared with the forward sensitivity simulations of the Models-3/CMAQ system. The two modeling approaches are highly comparable in their assessments of atmospheric pollution control schemes for critical emission regions, but the adjoint method has higher computational efficiency than the forward sensitivity method. The results also imply that critical regional emissions reduction could be more efficient than individual peak emission control for improving regional $PM_{2.5}$ air quality.

# 1. Introduction

The application of adjoint theory to atmospheric chemistry models can enable the efficient calculation of the sensitivities of a few variables or metrics with respect to a large number of input parameters (Marchuk, 1974; Sandu et al., 2005; Hakami et al., 2007). Classic atmospheric chemistry models use inputs of emission sources to output the spatial–temporal variation of pollutants, thus is source-oriented. By contrast, adjoint models are receptor-oriented, for they use the gradients of the cost function to model variables (usually pollutant concentrations) as inputs and output the spatial–temporal variations of the sensitivities of the cost function to emissions (Errico, 1997; Carmichael et al., 2008). Therefore, in concentration–source sensitivity analysis, the adjoint method is more computationally efficient than others, such as the traditional finite difference method, which requires repeated input perturbations and result comparisons (Wang et al., 2015). Moreover, the finite difference approach changes the state of the modeled atmosphere and inevitably incurs truncation and cancellation errors (Constantin and Barrett, 2014). When calculating gradients, the adjoint model integrates under certain atmospheric conditions; thus, it can provide exact sensitivities. Although the adjoint approach is not strictly a method used for source apportionment because it provides merely tangent linear derivatives (gradients) that are likely to be valid over only a limited range of values for the parameters (emissions), it does provide valuable information about the dependence of aerosol concentrations on emissions (Henze et al., 2007 and 2009; Zhang et al., 2015). If we set the cost function as the pollutant concentration over a region at a point in time (or during a time period), the adjoint sensitivity approach can detect critical emission sources in detail and reveal the changes in concentration due to perturbations in emission sources.

Beijing is a rapidly growing economic center and a densely populated metropolis whose recent $PM_{2.5}$ pollution problems have garnered considerable attention (Zhang et al., 2016; Sun et al., 2014; Guo et al., 2010; Wu et al., 2015). $PM_{2.5}$ pollution in Beijing is significantly influenced by the regional transport of pollutants from its environs. As such, the joint control over effective air pollution emission sources has been promoted. Research using approaches such as the flux calculation method (An et al., 2007), the back-trajectory model (Zhai et al., 2016), and observation analysis (Li et al., 2016), has revealed that southerly winds almost always promote high $PM_{2.5}$ conditions in Beijing. Studies have also indicated that more than 50% of $PM_{2.5}$ pollutants originate in surrounding provinces and cities, including southern Hebei, Tianjin, eastern Shanxi, and Shandong Provinces (Jiang et al., 2015; Gao et al., 2016). Studies have also shown that joint regional air pollution management control can be more cost-effective (Wu et al., 2015) and that joint control schemes in critical source zones (detected by a back-trajectory model) prior to unfavorable meteorological conditions can help reduce costs and improve efficiency (Zhai et al., 2016). The above studies either determined pollution pathways through meteorological analysis or analyzed air pollutant concentration sensitivities for a limited group of emission sources. If air pollution can be spatially and temporally traced back to its emission sources, decision-making regarding air pollution management can be better addressed.

Unlike back-trajectory approaches or statistical factor analysis, the adjoint approach accounts for chemical and physical processes combined with transport; thus, it efficiently estimates the incremental influence of specific sources on air quality (Henze et al., 2009). Recently, An et al. (2016) developed the aerosol adjoint module of the atmospheric chemical modeling system GRAPES-CUACE (the Global–Regional Assimilation and Prediction System coupled with the China Meteorological Administration Unified Atmospheric Chemistry Environment) and estimated the average black carbon (BC) concentrations over Beijing at the highest concentration time with respect to BC amounts emitted over the Beijing–Tianjin–Hebei region. They also indicated the effectiveness of controlling the most influential regions during critical time intervals, as detected by the adjoint sensitivity analysis. Zhang et al. (2015) attributed the sources of Beijing's $PM_{2.5}$ by using the GEOS-Chem adjoint model and summarized that residential (49.8%) and industrial sources (26.5%) are the largest contributors. They further noted that 45%–53% of $PM_{2.5}$ pollutants in Beijing and Tianjin are from local sources, whereas the Hebei Province sources contribute approximately 26%. Both Zhang et al. (2015) and An et al. (2016) demonstrated the high efficiency and accuracy of the atmospheric chemistry adjoint model in identifying Beijing air pollution sources.

In this study, we apply the newly developed GRAPES-CUACE aerosol adjoint model (An et al., 2016) to track the sensitive primary emission sources of a high $PM_{2.5}$ episode that occurred in Beijing in November 2012. The two $PM_{2.5}$ concentration peaks that occurred were set as the cost functions. By detecting the primary emission sources of these two hourly $PM_{2.5}$ peaks, our work advances the understanding of the impacts of emission sources by providing detailed insights into the spatial and temporal variability of emission source contributions from each of the surrounding provinces and from local and environs transports. We then set the average $PM_{2.5}$ concentration from November 21 as the cost function and compared the adjoint model results with the Models-3/CMAQ assessments (Zhai et al., 2016). Furthermore, we also compared emission source impacts on the Beijing $PM_{2.5}$ concentration peak from zones with maximum adjoint sensitivities and emission-intensive zones. This study explores the capability of the GRAPES-CUACE aerosol adjoint model to simulate detailed concentration–source relationships and provide guidance for flexible environmental control policy.

## 2.    Synoptic analysis of the pollution episode

Atmospheric stability and humidity over the mid-eastern region of China from the 19[th] to the 22[nd] of November 2012 were analyzed in combination with the results of the Meteorological Information Comprehensive Analysis Processing System, the sounding stratification and dew point-pressure curves (temperature-logarithmic pressure diagrams) from Nanjiao Station (Fig. 1left) in Beijing (Fig. 2), and the flow field pattern. Meanwhile, the formation of two pollution peaks at dawn and at night on November 21, 2012, was also qualitatively analyzed. During the period between the 19[th] and 20[th] of November, Beijing was under the influence of a low-pressure system situated between two high pressures. During the daytime, southerly winds prevailed below 925 and 1000 hPa, and the relative humidity increased during this time period. During the nighttime,

southerly winds shifted to northeasterly and easterly winds, thus transporting pollutants, together with water vapor, to Beijing. In this same time period, thermal inversions occurred below 850 hPa. The above analysis reveals that the accumulation of $PM_{2.5}$ concentrations was tightly connected with southerly winds during the daytime and easterly winds at night.

During the daytime on the 21[st] of November, the Beijing–Tianjin–Hebei area was located at the bottom of a high-pressure system, with easterly winds prevailing in the 850 hPa layer. The thermal inversion remained, and the relative humidity continued to increase. The mid-southern Hebei Province was influenced by a mass of cold air controlled by northerly winds, whereas Beijing was mainly under the influence of an easterly wind that promoted pollutant convergence in front of the Taihang Mountains and carried abundant water vapor, which accelerated the hygroscopic growth of local particles. It can be concluded, that the pollution peak on the night of November 21[st] was not only the result of the accumulation of pollutants during the previous 2 d but also the result of the hygroscopic growth of local particles and the convergence of pollutants caused by daytime easterly winds. According to prior research (Chen et al., 2016; Li et al., 2016), this event was typical of a synoptic episode that gradually generates air pollution over Beijing until a sudden and significant improvement in air quality due to strong winds. This is also the same episode that was analyzed by Zhai et al. (2016), thus facilitating further comparisons.

## 3.    Methods

### 3.1.  Concepts of the adjoint sensitivity analysis

Sensitivity analysis plays an important role in atmospheric environmental research. Understanding the impacts of emissions on pollutant concentrations is helpful for the development of effective air pollution control strategies. The adjoint model is efficient in calculating the sensitivity of an cost function to any model variable at any time step. Figure 3 shows the schematic diagrams of the forward atmospheric chemistry model and the adjoint model. The atmospheric chemistry model takes emissions (S: $S_1$, $S_2$, …, $S_n$, …, $S_N$) as inputs and outputs pollutant concentrations (C: $C_1$, $C_2$, …, $C_m$, …, $C_M$) through forward integration. Any emission source ($S_n$) might have an influence on the concentration at any receptor site ($C_m$). A pair of emission source sensitivity tests using the traditional source-oriented finite difference method can determine the contribution of an emission source (or a combined group of emission sources) to the pollution level at any receptor site. Therefore, with N emission sources and M receptors in total, the contribution from each of the N emission sources to each of the M receptors (an N × M matrix) can be obtained through N + 1 iterations of forward integration (one base simulation included). The receptor-oriented adjoint model is complementary to the forward model. The sensitivity map of a scalar function of pollutant concentration (the cost function) to every emission source (N × 1 matrix) can be obtained by performing one backward adjoint integration (Sandu, 2005; An et al., 2016; Zhai, 2015), with the above-mentioned N × M matrix

requiring M iterations of the adjoint integration. Theoretically, the N × M matrices resulting from the forward and backward methods are the same within a small perturbation (Marchuk, 1986), considering the nonlinearity of $PM_{2.5}$ formation.

Adjoint sensitivities are the tangent linear derivatives (gradients) of the cost function to model parameters (emissions) and are likely to be valid over only a limited range of values for each parameter (Henze et al., 2007 and 2009). In this study, the GRAPES-CUACE aerosol adjoint model considered only primary $PM_{2.5}$ (explained in Section 3.2), and the primary $PM_{2.5}$ emission sources and $PM_{2.5}$ concentrations had an approximately linear relationship (see Fig. S1 in the supplement). Given the linear relationship between the concentration of $PM_{2.5}$ and its primary emission sources, the magnitude of perturbations did not influence the representative of the adjoint sensitivities when comparing the contributing proportions of emission sources from different regions. However, if the adjoint sensitivities are used to represent the absolute emission source contributions, errors will increase with an increase in perturbations. In Fig. S1, we can see that the adjoint sensitivity results are similar to the finite difference results, and the difference between the adjoint sensitivity results and the finite difference results grows with the increase of emissions reduction ratios (the blue line with circular symbols and the red line with triangle symbols are close, particularly when the X-axis are within 30%); therefore, the adjoint sensitivity coefficients are likely to be representative over $PM_{2.5}$ primary emission reduction ratios from 5% to 90% or at least over a modest range of emission perturbations commensurate with typical emission abatement strategies (10–30%). All in all, an atmospheric chemistry model is suitable for simulating air pollution processes, whereas an adjoint model is efficient in quantifying receptor–source relationships.

The adjoint model can calculate the sensitivity of the cost function (J) to any emission source ($S_n$), as denoted by $\partial J/\partial S_n$. If we compare a group of uniformly distributed emission sources, larger $\partial J/\partial S_n$ values indicate the greater influence of $S_n$ on J. However, emission intensities are obviously not uniform across urban and rural areas, and seasonal and diurnal changes add even more nonuniformity. Furthermore, the emissions of different species of pollutants may have different units and may differ in their order of magnitude. Under these circumstances, the relative contribution of each emission source cannot be determined only by calculating the gradient $\partial J/\partial S_n$. Therefore, we define the sensitivity coefficients in this study as $(\partial J/\partial S_n) \cdot S_n$, which shares the same unit as the cost function and reflects the absolute changes in the cost function due to perturbations in emission sources; this definition makes the contrast between emission sources more convenient.

## 3.2. Model description

The GRAPES-CUACE is an online coupled atmospheric chemistry modeling system (Wang et al., 2009; Zhou et al., 2012; Jiang et al., 2015) developed by the CMA. GRAPES-Meso is a regional meteorological model (Xue et al., 2008) within GRAPES-CUACE, and CUACE is an atmospheric chemistry modeling system independent of meteorological and climate models (Gong et al., 2009). The CUACE system adopted the Canadian Aerosol Module (CAM) (Gong et al., 2003), a size-segregated multi-component aerosol algorithm, as its aerosol module and the second-generation Regional Acid

Deposition Model (RADM II) (Stockwell et al., 1990) as its gaseous chemistry model. CAM contains computations for numerous major aerosol processes in the atmosphere: generation, hygroscopic growth, coagulation, nucleation, condensation, dry deposition/sedimentation, below-cloud scavenging, aerosol activation, and chemical transformation of sulfur species in clear air and in clouds (Gong et al., 2003), which is coherently integrated with the gaseous chemistry component in CUACE. Given that the nitrates and ammonium formed through gaseous oxidation are unstable and prone to further decomposition back to their precursors, CUACE adopts ISORROPIA to calculate the thermodynamic equilibrium between them and their gas precursors (Zhou et al., 2012). The CUACE system is compatible with various kinds of meteorological models and can be used as a common platform for atmospheric constituent calculation.

The GRAPES-CUACE aerosol adjoint model was developed by applying adjoint theory to the GRAPES-CUACE modeling system. The current version of the adjoint model includes the adjoint of CAM (Gong et al., 2003), the adjoint of the three interface programs that pass meteorological variable values from GRAPES-Meso to chemical processes in CUACE, and the adjoint of the aerosol transport processes. Considering that the adjoint of the gaseous chemistry (RADM II) and the adjoint of the thermodynamic equilibrium (ISORROPIA) processes are not included in the GRAPES-CUACE aerosol adjoint model, the GRAPES-CUACE aerosol adjoint model is capable of simulating sensitivities of the cost function to primary $PM_{2.5}$ sources. Hence $S_n$ defined in section 3.1 refers to primary $PM_{2.5}$ sources. After the tangent linear model (TLM) and the adjoint model are built (the adjoint model is a concomitant of the TLM), they are divided into smaller sections and tested separately before the assembled TLM and the adjoint model are confirmed valid. The details of the adjoint verification can be found in An et al. (2016).

Figure 4 shows the operational processes used in this study. To ensure that the forward and backward models were in the same chemical state, the forward GRAPES-CUACE model was first integrated to save the model state variables (concentrations) in checkpoint files at the beginning of each external time step (Sandu et al., 2005; Henze et al., 2007). These saved variables were then inputted at each checkpoint during the backward adjoint integration. To handle intermediate variables, this study adopted both recalculation and stack storage (PUSH & POP) schemes. Details about the construction, framework, and operational flowchart of the GRAPES-CUACE aerosol adjoint model are discussed in An et al. (2016).

### 3.3. Model setup, data, and validation

The simulated domain in this study covered northeast China (105°E–125°E, 32.25°N–42.25°N) (Fig. 1), which included 41 × 23 simulation grid cells with 31 vertical layers at the resolution of 0.5° × 0.5°. The model was integrated at a time step of 300 s. The National Centers for Environmental Prediction Final Analysis dataset was used to define the initial meteorological field and the meteorological boundary conditions. The initial and boundary values for $O_3$ and OH were taken from climatic means and zeros for each aerosol species during the first run; thereafter, the daily initial values of all chemical species were determined by the 24 h forecast made by the previous day's simulation. To eliminate the discrepancy between the idealized

initial concentration field and the real concentration field, the simulation was started at 20:00 Beijing LT (GMT+8) on November 10, 2012, with the analysis period running from 20:00 LT on November 17, 2012, to 19:00 LT on November 22, 2012.

This study used hourly gridded off-line emission sources processed by the SMOKE module, which is based on statistical data of anthropogenic emissions reported from government agencies for 2007. Anthropogenic emissions include primary $PM_{2.5}$ and pollutant gases (Cao et al., 2011). Emission source types included biomass combustion, residences, power generation, industry, transportation, livestock and poultry breeding, fertilizer use, waste disposal, solvent use, and light industrial product manufacturing (Cao et al., 2011). Furthermore, natural sea salt and natural sand/dust emissions were also calculated in the model.

Figure 5 illustrates the gridded distribution of the overall primary $PM_{2.5}$ sources. Figure 6 shows the hourly variability of the overall $PM_{2.5}$ sources in Beijing. In Fig. 5, there are four intensive source zones over Beijing and its surrounding provinces: 1) southern Beijing and Tianjin (TJ), 2) southern Hebei (HB), 3) middle Shanxi (SX), and 4) north central Shandong (SD). Meanwhile, a secondary intensive source zone was observed over northern SX. In Fig. 6, it is noted that the overall primary $PM_{2.5}$ source emission intensity decreased to its lowest level at 05:00 LT. Thereafter, emission intensity began to increase and remained high from 11:00 to 19:00 LT, with a little trough at 14:00 LT.

The observation data includes meteorological elements (2 m temperature and 10 m wind speed) and $PM_{2.5}$ concentrations. The meteorological data were collected from the Nanjiao (NJ: 116.47°E, 39.8°N), Haidian (HD: 116.28°E, 39.93°N) and Shangdianzi (SDZ: 117.12°E, 40.65°N) stations. The NJ and HD stations are representative urban observatory stations and the SDZ station is a typical background station. These three stations are part of the measurement network run by the Beijing Meteorology Bureau and uses standard measurement equipment and methods. $PM_{2.5}$ measurements used in this study were obtained from the observation stations of the Chinese Research Academy of Environmental Sciences (CRAES: 116.39°E, 40.03°N), as well as of Guanyuan (GY: 116.34°E, 39.93°N) and Dingling (DL: 116.22°E, 40.29°N). The CRAES station is located in the northwest Chaoyang District at the Chinese Academy of Environmental Sciences, and the GY station is located in Xicheng District. Both the CRAES station and the GY station are representative urban observation stations in Beijing. The DL station is located in the relatively clean Changping District in northern Beijing and provides background values for observed $PM_{2.5}$ concentrations (Fig. 1).

The reliability of the GRAPES-CUACE modeling system is evaluated in terms of both meteorological and chemical simulations. Figure 7 shows the hourly variations of the observed and simulated 2 m temperature (T2m) and 10 m wind speed (WS10m), and Table 1 lists the corresponding statistical parameters. The correlation coefficients (Rs) between the observed and simulated hourly T2m are 0.77, 0.75 and 0.74, passing the 99% confidence level with root mean square error (RMSE) values of 1.5, 1.6 and 1.7 °C, respectively at observatory sites NJ, HD and SDZ. Mean Biase (MB) values for the

T2m demonstrate a slight underestimate in NJ (-0.1 °C) and HD (-0.3 °C), and overestimate in SDZ (0.8 °C). The variations of the WS10m are generally captured by the model with Rs of 0.70, 0.73 and 0.46, and with RMSEs of 1.4, 1.5 and 1.8 m s$^{-1}$ at NJ, HD and SDZ stations respectively (passed the 99% confidence level). Overall, the GRAPES-Meso could reasonably reproduce the observed meteorology.

Figure 8a–6c show the observed and simulated hourly $PM_{2.5}$ concentration curves from 20:00 LT on November 17 to 19:00 LT on November 22 at the CRAES, GY, and DL observational stations, and Table 2 lists the statistical parameters. Figure 8a–6c reveal that the results of the GRAPES-CUACE modeling system correspond well with the synoptic analysis of the pollution episode. The modeling system was able to reproduce the $PM_{2.5}$ accumulation processes observed from the 19[th] to 21[st] of November in Beijing and captured the two $PM_{2.5}$ hourly concentration peaks during the dawn and night of November 21, as well as the trough during the afternoon on November 21 at the CRAES, GY, and DL stations, with correlation coefficients (Rs) of 0.87, 0.91, and 0.69, respectively (Table 2). However, the model overestimated $PM_{2.5}$ concentration values over the period with normalized mean biases (NMBs) of 57.2%, 108.1%, and 10.7% at the CRAES, GY, and DL stations, respectively. The overestimation was also reflected in the positive mean bias (MB) and mean fractional bias (MFB) values. For the CRAES, GY, and DL stations, the MFBs were 53.6%, 65.2%, and 15.6%, respectively, and the corresponding mean fractional errors (MFEs) were 60.1%, 68.3%, and 39.6%, respectively. MFEs and MFBs are all within the criteria proposed by Boylan and Russel (2006)—model performance criteria are met when the MFE and MFB are less than or equal to approximately +75% and ±60%, respectively—except for the MFB at GY, which is a little high. Secondary aerosol formations are important processes in atmospheric physics and chemistry and have large uncertainties, according to the current understanding on the atmospheric environment. The lack of heterogeneous chemical reactions (Wang et al., 2016; Cheng et al., 2016; Guo et al., 2014; Zhang et al., 2015) in the forward GRAPES-CUACE model could be a factor contributing to the modeling uncertainties in this study. Generally, the three factors controlling the discrepancies in air quality modeling are as follows: 1) air pollutant emissions, 2) physical and chemical processes in the atmosphere, and 3) meteorology, particularly in the boundary layer (An et al., 2013; Cheng et al., 2016; Wang et al., 2015a; Wang et al., 2016). The overestimation of $PM_{2.5}$ in this study might be attributed to the uncertainties of these three factors in the model. Prior studies (Zhou et al., 2012; Wang et al., 2015a; Wang et al., 2015b; Jiang et al., 2015) have demonstrated the stable simulation performance of the GRAPES-CUACE modeling system in reproducing air pollution levels and variation trends over northeast China. Above all, the following analysis mainly focuses on the variations and the contributing proportions of emission sources over different regions. Therefore, adjoint sensitivity analysis was not significantly affected by the overestimation of $PM_{2.5}$, and these modeling results can be considered reliable.

## 4. Results

### 4.1. Simulated haze episode and cost function

Figure 9 shows the simulated surface PM$_{2.5}$ concentrations and the wind field variations from 17:00 LT on November 19 to 11:00 LT on November 22. It can be seen that the simulation results are consistent with the qualitative weather analysis of this time period. From the 19[th] to the 20[th] of November, PM$_{2.5}$ accumulated in Beijing under the influence of a convergent wind field pattern: a southerly wind field to the south, an easterly wind field to the east, and a westerly wind field to the west.

From 5:00 LT to 11:00 LT on November 21, PM$_{2.5}$ concentrations exceeded 550 μg m$^{-3}$ over southern Beijing, south-central Hebei, and northwest Tianjin. After this peak, PM$_{2.5}$ concentrations over Beijing, south-central Hebei, and Tianjin decreased to a trough in the afternoon, before rising again to above 550 μg m$^{-3}$ at 23:00 LT. The decrease in PM$_{2.5}$ concentrations from the morning to the afternoon is typical for Beijing and resulted mainly from diurnal variation of the planetary boundary layer, with vertical mixing after sunrise effectively diluting the pollutants (Zhao et al., 2009; Liu et al., 2015; Tang et al., 2016).

The concentration peak at 23:00 LT was driven by the influence of the easterly winds, which caused pollutant convergence against the Taihang Mountains and carried abundant water vapor that promoted local hygroscopic growth. Thereafter, during the daytime on November 22, a notable northwesterly wind dispersed pollutants in Beijing, thus ending this pollution episode.

The municipality of Beijing (covering both rural and urban Beijing) experienced two hourly PM$_{2.5}$ concentration peaks at

5:00 LT and 23:00 LT on November 21 (Fig. 8d), similar to those observed at the three observation stations. These peaks resulted in the observed high daily average PM$_{2.5}$ concentration on November 21, which was analyzed in previous research (Zhai et al., 2016). To analyze the critical emission sources of the two hourly PM$_{2.5}$ concentration peaks, we took advantage of the adjoint model for simulating concentration–emission relationships and defined two cost functions as the hourly mean PM$_{2.5}$ concentrations over Beijing at (i) 5:00 LT and (ii) 23:00 LT on November 21. To demonstrate the reliability and

efficiency of the GRAPES-CUACE aerosol adjoint model to provide guidance toward effective and flexible air quality control designs, a third cost function was defined as (iii) the average PM$_{2.5}$ concentration over Beijing on November 21. Subsequently, comparisons between results from the GRAPES-CUACE aerosol adjoint model and the Models-3/CMAQ assessments (Zhai et al., 2016) were made.

### 4.2. Spatial distribution of primary PM$_{2.5}$ emission source sensitivity coefficients

Figure 10 illustrates the distribution of time-integrated sensitivity coefficients to emission sources for the two concentration peaks in the hourly PM$_{2.5}$ in Beijing. The sensitivity coefficients of the cost function to emission sources connected pollutants with emissions and revealed the incremental impacts of emissions on peak PM$_{2.5}$ concentrations. A larger sensitivity coefficient value corresponds to its greater influence on the cost function, J. For example, the largest sensitivity

coefficient in Figure 10d was in the cell that includes Daxing District, with a value of 22.4 µg m$^{-3}$. This indicates that emissions stemming from this area had the greatest influence on the peak concentration when integrated over 72 h. If emissions were reduced within a small range, the decrease in PM$_{2.5}$ concentrations should be linear. For example, if emissions from this cell were reduced by N% from 05:00 LT on November 18 to 05:00 LT on November 21, the target PM$_{2.5}$ concentration would decrease by N%*22.4 µg m$^{-3}$.

When looking at the accumulation along an inverse time sequence, as shown in Figs. 10a-h, the more influential regions (regions with relatively larger sensitivity coefficients) extended from local Beijing (the target region that covers the entire Beijing Municipality) to its surrounding provinces. This phenomenon reflected that the PM$_{2.5}$ pollution episode in Beijing was not only the result of local emissions but also the result of emissions from surrounding regions, including Hebei Province, Tianjin, and even Shanxi and Shandong Provinces. Emissions from the surrounding areas were continuously transported to Beijing 2 to 3 d ahead of the peak pollution day, thus leading to the observed increase in Beijing's air pollution concentration.

There are differences in the variations in the more sensitive emission regions of these two PM$_{2.5}$ concentration peaks. First, by comparing the 12 h cumulative sensitivity coefficients distribution in Figs. 10b and 10f, we can see that emissions to the southwest of Beijing already had a clear influence on the 05:00 LT November 21 PM$_{2.5}$ concentration peak (Fig. 10b). However, for the 23:00 LT November 21 PM$_{2.5}$ concentration peak, the influential emission sources were still concentrated over Beijing Municipality (Fig. 10f), with only a small fraction of influential emissions coming from the east and south of Beijing. This is due to the southwesterly airstream positioned to the southwest of Beijing from 23:00 LT on November 20 to 05:00 LT on November 21, and the southeasterly water vapor imported during the afternoon and night of November 21, which caused the moisture–absorption growth of local particles and brought pollutants from Tianjin.

Second, it can be seen from the distributions of the 24 (Figs. 10c and 10g) and 72 h (Figs. 10d and 10h) cumulative sensitivity coefficients that sensitivity coefficients both in and around Beijing had relatively large values, thus indicating that both of these PM$_{2.5}$ concentration peaks were influenced by local and surrounding emissions. However, the most influential emission regions differed between the two PM$_{2.5}$ concentration peaks. For the first PM$_{2.5}$ concentration peak, the key 24 h source regions (Fig. 10c) were distributed over Beijing and to the west and south of Beijing. The key 72 h source regions (Fig. 10g) were to the northeast in Shanxi Province. However, for the second PM$_{2.5}$ concentration peak, the key 24 h source regions were mainly located to the south of Beijing, whereas the key 72 h source regions were to the west of Beijing (Shanxi Province) (Fig. 10h) and covered a smaller area than that for the first PM$_{2.5}$ concentration peak (Fig. 10d).

The results of these simulations show that the variation in the distribution of the sensitivity coefficients, the meteorological conditions, and the pollution evolution processes correspond with each other very well. This indicates that the

GRAPES-CUACE aerosol adjoint model is capable of estimating the sensitivity of concentrations to emission sources by propagating a perturbation in concentration backward in time by incorporating meteorological and chemical processes.

### 4.3. Influence of local and surrounding emission sources on peak PM$_{2.5}$ concentrations

Figure 11 illustrates the hourly instantaneous sensitivity coefficients to local Beijing (the target region that covers the entire Beijing Municipality), its surrounding emission sources (emissions from Hebei, Tianjin, Shandong, and Shanxi Provinces) (Figs. 11a and 11b), and their corresponding time-integrated series (Figs. 11c and 11d). The magnitudes of the sensitivity coefficients reflect the incremental influence of local and surrounding emissions to the objective PM$_{2.5}$ peaks. It can be seen that the instantaneous sensitivity coefficients of the PM$_{2.5}$ concentration peaks to local (red closed squares) and surrounding (red open squares) emissions increased to their maximal points before showing a decreasing tendency. However, detailed comparisons of the hourly contribution revealed significant differences between the local and surrounding emissions.

When studying Figs. 11a and 11b along a reversed time sequence, the local emission sensitivity coefficient maximums (red closed squares) and the PM$_{2.5}$ concentration peaks (black closed circles) appeared at almost the same time, with the latter delayed by 1 to 2 h. This indicates that local emissions released 1 to 2 h ahead of the PM$_{2.5}$ peak values were the main contributors to the peak pollution concentrations. After the sensitivity coefficient reached a maximum, local emission sensitivity coefficients decreased sharply to minimal values at 14 h (for the 05:00 LT PM$_{2.5}$ peak) or 19 h (for the 23:00 LT PM$_{2.5}$ peak) ahead of the pollution peak and remained low. This revealed that PM$_{2.5}$ generated from local emissions was transported away from Beijing after about 14–19 h.

By contrast, maximal sensitivity coefficients of the surrounding emissions (red open squares) occurred 7–12 h ahead of the PM$_{2.5}$ concentration peaks (Figs. 11a and 11b), thus indicating a 7 to 12 h delay in the arrival of emissions from surrounding areas to Beijing. Similar to the backward integration, sensitivity coefficients showed overall decreasing trends with periodic fluctuations. For the first PM$_{2.5}$ concentration peak (05:00 LT on November 21), three maximal contributions from surrounding areas (Fig. 11a) appeared along the reversed time sequence at 17:00 LT on November 20 (12 h ahead of the target time), 1:00 LT on November 20 (28 h ahead of the target time), and 4:00 LT on November 19 (49 h ahead of the target time). The first time-reversed relative maximal sensitivity coefficient of 7.5 µg m$^{-3}$ was noted at 17:00 LT on November 20, whereas the second and the third time-reversed relative maximal sensitivity coefficients of 5.2 and 1.5 µg m$^{-3}$ were observed at 1:00 LT on November 20 and 4:00 LT on November 19, respectively. For the second PM$_{2.5}$ concentration peak (23:00 LT on November 21) (Fig. 11b), the relative maximal contributions from surrounding areas (red open squares) appeared at 16:00 LT on November 21 (7 h ahead of the objective time), at 20:00 LT on November 20 (27 h ahead of the objective time), at 23:00 LT on November 19 (48 h ahead of the objective time), and at 3:00 LT on November 19 (68 h ahead of the objective time); their corresponding sensitivity coefficients were 5.3, 5.4, 2.6, and 0.9 µg m$^{-3}$, respectively. It is worth noting that sensitivity coefficients maximal points for the 23:00 LT PM$_{2.5}$ peak appeared at time points similar to those of the sensitivity

coefficients maximal points for the 05:00 LT $PM_{2.5}$ peak. The sensitivity coefficients around the second maximal contribution, approximately from 17:00 LT on November 20 to 0:00 LT on November 21, remained at a relatively large value (about 4.7 to 5.4 μg m$^{-3}$), even slightly larger than that of the first maximal sensitivity coefficient. This is because the second $PM_{2.5}$ concentration peak was the result of cumulative increases based on the first high $PM_{2.5}$ concentration peak; therefore, emissions from the surrounding areas from the night of November 20 to early in the morning on November 21 also had a large influence on the second $PM_{2.5}$ concentration peak, almost slightly rivaling the influence of the later emissions sensitivity peak.

On the basis of Fig. 11, we can also see that for both $PM_{2.5}$ concentration peaks, the dominant emission source areas shifted from the local to the surroundings areas over the backward time sequence (Figs. 11c and 11d). For the first $PM_{2.5}$ concentration peak (05:00 LT on November 21) (Fig. 11c), the cumulative local emission sensitivity coefficients (red closed squares) were larger than the surrounding emission sensitivity coefficients (red open squares) between 12:00 LT on November 20 and 05:00 LT on November 21 (lasted for 17 h), thus indicating that local emissions dominated during this 17 h time period. For the second $PM_{2.5}$ concentration peak (23:00 LT on November 21) (Fig. 11d), local emissions dominated from 21:00 LT on November 20 to 23:00 LT on November 21, which lasted for 26 h (9 h longer than that of the first $PM_{2.5}$ peak pollution period). This phenomenon indicates the tiny effect of emission transport processes on November 21 and that the increase in $PM_{2.5}$ concentrations on November 21 was mainly due to local source generation. This reinforces the importance of the impact of emissions from surrounding regions on the accumulation seen in the first $PM_{2.5}$ concentration peak.

### 4.4. Impact of emission sources from different provinces around Beijing to peak $PM_{2.5}$ concentrations

The emission sensitivity coefficients were then divided into different provinces around Beijing to investigate their influence on the $PM_{2.5}$ concentration peaks over Beijing Municipality. Figure 12 illustrates the hourly instantaneous sensitivity coefficients to emission sources from Beijing Municipality (BJ), Hebei Province (HB), Tianjin city (TJ), Shanxi Province (SX), and Shandong Province (SD) (Figs. 12a and 12b), their corresponding time-integrated series (Figs. 12c and 12d), and the overall contribution proportions of the emission sources from each province to the $PM_{2.5}$ concentration peaks (Figs. 12e and 12f). As shown in Fig. 12, the impacts of emission sources from BJ, HB, TJ, SX, and SD on BJ $PM_{2.5}$ concentration peaks are quite different in both variability and magnitude.

For the $PM_{2.5}$ concentration peak occurring at 05:00 LT on November 21, emission sources from HB contributed the most among surrounding provinces, and the variation in HB's hourly sensitivity coefficients showed consistent periodic fluctuations with that of surrounding emissions. Three maximal points of the HB hourly sensitivity coefficients of variation occurred at the same time as that of surrounding emission sources. Corresponding sensitivity coefficients were 5.3, 3.2, and 0.8 μg m$^{-3}$, respectively (Fig. 12a). The largest influential time period for emissions from TJ appeared 13 h ahead of the

objective time (at 16:00 LT on November 20), followed by an obvious secondary maximal point that appeared 24 h ahead of the objective time (at 05:00 LT on November 20). Sensitivity coefficients from SX showed a small peak (approximately 0.7 $\mu g\ m^{-3}$) 9 h ahead of the objective time (at 20:00 LT on November 20), which was caused by a secondary intensive emission zone in northern SX that was relatively close to BJ (Fig. 5). As intensive emission sources in SX and SD are far from BJ (Fig. 5), it took 33–36 h for SX and SD emissions to reach BJ.

It is worth noting that, except for the maximal sensitivity coefficients of HB and TJ observed at 16:00 LT on November 21 (7 h ahead of 23:00 LT on November 21), prior sensitivity coefficient maximal points for the $PM_{2.5}$ concentration peak observed at 23:00 LT on November 21 appeared at the same time as the maximal points of sensitivity coefficients when the $PM_{2.5}$ concentration peak observed at 05:00 LT on November 21 was set as the cost function. For example, for both $PM_{2.5}$ concentration peaks, sensitivity coefficients of TJ emission sources reached a maximal point at 16:00 LT on November 20, and SX emission source sensitivity coefficients in turn showed two maximal points at 20:00 LT on November 20 and at 20:00 LT on November 19. The situations at HB and SD are similar: even when maximal points do not appear at the exact same time, high value periods are consistent for the two cost functions. The above phenomenon again revealed that the $PM_{2.5}$ concentration peak observed at 23:00 LT on November 21 was cumulative on the basis of the $PM_{2.5}$ concentration peak observed at 05:00 LT on November 21 and that if the $PM_{2.5}$ concentration peak at 05:00 LT on November 21 can be effectively reduced, the $PM_{2.5}$ concentration peak at 23:00 LT on November 21 can be reduced accordingly, thus decreasing the overall $PM_{2.5}$ concentrations on November 21. These results also reflected the advantage of the adjoint model in detecting temporal–spatial sensitive emission sources in detail.

Figures 12c and 12d show that along the backward time sequence, the time-integrated sensitivity coefficients of HB continuously rose after the time-integrated sensitivity coefficients of other provinces were prone to remain constant. At around 02:00 LT to 03:00 LT on November 20, the time-cumulated emissions influence from HB exceeded that from local BJ emissions for both $PM_{2.5}$ concentration peaks, thus reflecting that emissions from HB played a leading role in pollutant accumulation for the first BJ $PM_{2.5}$ concentration peak and that the influence of local emissions was dominant between the two $PM_{2.5}$ concentration peaks, that is, during the daytime on November 21.

The hourly sensitivity coefficients in Figs. 12a and 12b show that the impact of emission sources from Beijing and each surrounding province decreased to negligible values (close to zero) 72 h ahead of the objective time points. Meanwhile, corresponding time-integrated sensitivity coefficients in Figs. 12c and 12d also stopped increasing 72 h prior to the objective time points. Therefore, by integrating sensitivity coefficients 72 h ahead of the two $PM_{2.5}$ concentration peaks, we can obtain the overall contributing proportions of emission sources from each province to the BJ $PM_{2.5}$ concentration peaks (Figs. 12e and 12f). Among all provinces, HB has the largest impact on the two $PM_{2.5}$ concentration peaks, and the contribution of HB emissions to the first $PM_{2.5}$ concentration peak (43.6%) was greater than to the second $PM_{2.5}$ concentration peak (41.5%).

For the 05:00 LT $PM_{2.5}$ concentration peak, the second largest emission source contribution was from Beijing (31.2%), followed by SX (9.8%), TJ (9.8%), and SD (5.7%); for the 23:00 LT $PM_{2.5}$ concentration peak, the second largest emission source contribution was from Beijing (35.7%), followed by SX (8.1%), SD (8.0%), and TJ (6.7%).

From all the above analysis, we can conclude that joint management control of air pollution sources in Hebei Province, Tianjin City, and Shandong and Shanxi Provinces 2 to 3 d ahead of the first $PM_{2.5}$ concentration peak can effectively reduce $PM_{2.5}$ concentration accumulation resulting from the transport of pollutants, thus decreasing the BJ $PM_{2.5}$ concentration peaks.

### 4.5. Comparisons of the adjoint results with Models-3/CMAQ assessments

Prior research used a back-trajectory model, namely, FLEXPART, to locate sensitive emission regions of Yanqihu, Beijing, on November 2012. The study then used the Models-3/CMAQ modeling system to quantify the effects of emission reduction schemes at different ratios, during different time periods, and over different regions on the reduction of $PM_{2.5}$ concentrations on November 21 in Beijing (Zhai et al., 2016). On the basis of these results, we set the average $PM_{2.5}$ concentration over Beijing Municipality on November 21 as the cost function and compared the adjoint results with the Models-3/CMAQ assessments. Figure 13 illustrates the time-integrated sensitivity coefficient distributions when the Beijing average $PM_{2.5}$ concentration on November 21 was set as the cost function. The magnitudes of the sensitivity coefficients reflect the incremental influence of primary emission sources on the objective $PM_{2.5}$ concentrations. Similar to previous research (Zhai et al., 2016) that advocated the joint management control of emissions with the surrounding provinces 2 to 3 d ahead of the most polluted day, adjoint time-integrated sensitivity was intensified and extended during 48 to 72 h backward time integration.

To assess the adjoint sensitive source zone on decreasing $PM_{2.5}$ concentrations over Beijing and to compare the adjoint results with the Models-3/CMAQ assessments, we referred to the research by Zhai et al. (2016) and selected four emission regions: the overall Huabei region (HuaB), the sensitive Huabei region (HuaB-sens), the overall Beijing Municipality (BJ), and the sensitive Beijing region (BJ-sens) (Fig. 14). Grid cells with 72 h cumulative sensitivity coefficients larger than 3 μg $m^{-3}$ were included in the sensitive emission regions (HuaB-sens and BJ-sens), and grid cells with smaller sensitive values are outside the sensitive emission regions. Therefore, sensitive emission regions have relatively larger impact on the $PM_{2.5}$ peak concentrations than regions outside them. Here the HuaB-sens accounts for 10.2% of the area of HuaB and the BJ-sens accounts for 60.0% of the area of BJ, thus making them analogous to the regions defined by Zhai et al. (2016). In the work by Zhai et al. (2016), HuaB-sens accounted for 17.6% of the area of HuaB and BJ-sens accounted for 54.2% of the area of BJ. Furthermore, on the basis of the emission magnitudes (Fig. 5), we defined regions with emission intensities larger than $4.1 \times 10^{-7}$ g·$s^{-1}$ within HuaB as the "Emis-intensive" regions (Fig. 14). The Emis-intensive region has the same area as that of the HuaB-sens.

Table 3 lists the ratios of the time cumulative sensitivity coefficients to peak $PM_{2.5}$ concentrations (SC/PC) over the BJ, BJ-sens, HuaB, HuaB-sens, and Emis-intensive regions at three different time points: 0 (d0), 1 (d1), and 2 d (d2) in advance of the most polluted day. The SC/PC reflects the reduction ratios of peak $PM_{2.5}$ concentrations due to the absence of emissions over different regions and during different periods, that is, emission source contribution ratios to peak $PM_{2.5}$ concentrations. From Table 3, we can see that the adjoint model results are highly consistent with the Models-3/CMAQ system results (Zhai et al., 2016). The $PM_{2.5}$ concentrations on November 21 reflect an accumulated result from emissions released in the day or 2 d prior to the most polluted day rather than a simple result of emissions on November 21. For all the BJ, BJ-sens, HuaB, and HuaB-sens regions, emission contribution ratios grew from "d0" to "d2" ("d0," "d1," and "d2" are defined in the caption of Table 3), particularly from "d0" to "d1." The contribution ratios of emissions from BJ (and BJ-sens) and HuaB (and HuaB-sens) increased by 6.2% (5.8%) and 31.9% (18.9%) from "d0" to "d1," respectively. Thereafter, the contribution ratios again increased by 0.6% (0.5%) and 9.6% (3.6%), respectively, for emissions over BJ (or BJ-sens) and HuaB (or HuaB-sens) from "d1" to "d2." The above phenomenon also indicates that with the accumulation of time-reversed integration from 48 to 72 h prior to November 21, emission source contributions from HuaB (or HuaB-sens) to peak $PM_{2.5}$ concentrations increased more obviously, whereas emission source contributions from BJ (or BJ-sens) hardly increased at all. This can be explained by surrounding emissions being continuously transported to Beijing 2 to 3 d ahead of the most polluted day (Zhai et al., 2016).

Similar to the work in Models-3/CMAQ assessments, Table 4 shows comparisons of sensitive emission, full emission, and Emis-intense region source contribution effects and efficiencies to peak $PM_{2.5}$ concentrations. In Table 4, S/F(effect) in the BJ-sens column refers to the ratios of sensitivity coefficients over BJ-sens to sensitivity coefficients over BJ, and S/F(effect) in the HuaB-sens (or the Emis-intense) column refers to the ratios of sensitivity coefficients over HuaB-sens (or Emis-intense) to sensitivity coefficients over HuaB. Correspondingly, S/F(efficiency) refers to the ratios of sensitivity coefficients per unit area over BJ-sens (or over HuaB-sens and Emis-intense) to sensitivity coefficients per unit area over BJ (or over HuaB). Therefore, S/F(effect) and S/F(efficiency) reflect emission source reduction effects and reduction efficiency over critical (or emission intensive) regions. The implication of "d0," "d1," and "d2" results in Table 4 are the same as they are in Table 3. As shown in Table 4, the contribution efficiencies (contribution ratios per unit area) of emissions from the HuaB-sens and BJ-sens regions are significantly higher than those from the corresponding entire HuaB and BJ regions, respectively. Although BJ-sens covers only 60% of the area of the entire BJ, its contribution to the peak $PM_{2.5}$ concentrations is 86.6%–88.2% of that of the entire BJ. Its source contribution efficiency is 1.4 to 1.5 times that of BJ. Similarly, HuaB-sens covers only 10.2% of the area of the entire HuaB, but its contribution to the peak $PM_{2.5}$ concentrations is 61.0%–71.9% of that of the entire HuaB, and its source contribution efficiency is 6.0 to 7.0 times that of the entire HuaB (Table 4). Finally, emissions from HuaB-sens contribute much more than emissions only from BJ-sens, which supports joint management

control. Analogously, in the Models-3/CMAQ assessments, BJ-sens (or HuaB-sens) covers 54.2% (or 17.6) of the area of BJ (or HuaB), and its emissions reduction effect is 99.2%–100% (or 87.2%–93.7%) of that of the entire BJ (or HuaB), and its source contribution efficiency is 1.8 to 1.9 times (or 5.0 to 5.3 times) that of BJ (or HuaB).

We then compared emission source contribution ratios, effect, and efficiency from the HuaB-sens and the Emis-intense regions. As shown in Table 3 and Table 4, although the Emis-intense region has the same area as HuaB-sens, its SC/PC, S/F (effect), and S/F(efficiency) are all much smaller. The source contribution ratios to $PM_{2.5}$ concentrations on November 21 (SC/PC) from the "Emis-intense" regions are 9.7%, 17.6%, and 18.5% smaller, respectively, than those from HuaB-sens (Table 3), and the source contribution effect from the "Emis-intense" regions (S/F (effect)) are 37.9%, 30.7%, and 27.6% smaller, respectively, than the S/F (effect) of HuaB-sens, thus indicating that controlling air pollution sources from adjoint critical emission regions has better effects and higher efficiency than controlling emission sources from emission-intensive regions.

The computational loads of the adjoint simulation were much smaller than the comparable assessments made with the Models-3/CMAQ modeling (Zhai et al., 2016). For the adjoint simulation, one forward integration (for model state variables saving) and one backward adjoint integration can enable the determination of the influence of emissions from any source region during any time period to $PM_{2.5}$ concentration peaks. For the Models-3/CMAQ assessments, to compare the effects of emission reductions over two different time periods at two different ratios and over four different regions, 12 sensitivity tests with a control simulation are required. Although the deficiency of the adjoint analysis in this study is that we did not include $PM_{2.5}$ concentration precursor emission impacts, we find through comparison that the two modeling approaches are highly comparable in their assessments of atmospheric pollution control for critical emission regions. Overall, the adjoint sensitivities of peak $PM_{2.5}$ concentrations to primary $PM_{2.5}$ emissions using the GRAPES-CUACE aerosol adjoint model can provide valuable reference for evaluating emission impacts on pollutant concentrations and air quality control.

## 5. Conclusions

In this research, the GRAPES-CUACE aerosol adjoint model was applied to detect the pivotal emission sources of a November 2012 haze episode over Beijing, and the hourly peak $PM_{2.5}$ concentrations at 05:00 LT and 23:00 LT on November 21, 2012, were set as the cost functions. The peak $PM_{2.5}$ concentration contributions from local Beijing emissions and neighboring provinces were well compared. The adjoint model results corresponded well with the real weather analysis for this period and correctly described the spatial distribution of the most influential emission sources over time for both $PM_{2.5}$ concentration peaks. The 05:00 LT $PM_{2.5}$ concentration peak was mainly influenced by local Beijing emissions and the emissions from Hebei, Tianjin, and Shanxi because of the transmission of pollutants 2 to 3 d ahead of the peak time. The 23:00 LT $PM_{2.5}$ concentration peak was more sensitive to local Beijing emissions, and the regions to the south of Beijing in

Hebei Province, because of the accumulation from the first PM$_{2.5}$ concentration peak, local particle hygroscopic growth, and pollutants trapped against of the Taihang Mountains on November 21. The upstream Hebei province has the largest impact on both PM$_{2.5}$ concentration peaks, and the contribution of Hebei emissions to the first PM$_{2.5}$ concentration peak (43.6%) was greater than that to the second PM$_{2.5}$ concentration peak (41.5%). In Beijing, PM$_{2.5}$ concentration peaks responded to local emissions in 1 to 2 h, whereas surrounding emissions took 7 to 12 h to influence Beijing's air quality. The relationship between PM$_{2.5}$ and their primary emission sources is complicated by different weather conditions. Aerosol impacts on meteorological fields could be significant, which might further affect the aerosol pollution condition in the lower troposphere. Also, aerosol-cloud interactions might modify temperature and moisture profiles and precipitation (Wang et al., 2011), leading to potential feedback on the atmospheric chemistry. Moreover, climate change also has potential impacts on the pollution conditions in China (Wu et al., 2016). Further studies are required to investigate the relationship with adjoint sensitivities' representation of emission source contribution under different weather conditions.

We compared the adjoint results with Models-3/CMAQ assessments and found that the adjoint model results can provide evidence for all the conclusions supported by the Models-3/CMAQ assessments (Zhai et al., 2016). We then defined the "Emis-intense" region as an emission-intensive region within the Huabei region that has the same area as that of sensitive Huabei region (HuaB-sens) and compared its emission source contributions with those of HuaB-sens and HuaB. Overall, we concluded that narrowing the emission sources reduction scope to target critical source zones (zones detected by an adjoint model or a FLEXPART model), rather than emission-intensive regions, 2 to 3 d prior to unfavorable meteorological conditions can effectively decrease PM$_{2.5}$ concentrations and improve the efficiency of PM$_{2.5}$ reduction measures. Meanwhile, the adjoint simulation is far more computationally efficient than the assessments with Models-3/CMAQ modeling. The adjoint method is a powerful tool for simulating the relationship between emissions and concentrations, and it can be utilized to help improve flexible air quality control schemes. As we are now coupling the CB-IV mechanism in the GRAPES-CUACE forward model and embedding the CB-IV adjoint into the adjoint of GRAPES-CUACE, we will estimate sensitivities to both primary and precursor gaseous emission sources after this development.

*Acknowledgments.* This work was supported by the National Nature Science Foundation of China (Grant Nos. 41575151 and 91644223), the National Key R &D Program Pilot Projects of China (Grant No. 2016YFC0203304), and the Program for Postgraduate Research Innovation of Jiangsu Higher Education Institutions (Grant No. KYZZ16_0346).

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

Table 1 Performance statistics between observed and simulated meteorology

| | Nanjiao | | | | | Haidian | | | | | Shangdianzi | | | | |
|---|---|---|---|---|---|---|---|---|---|---|---|---|---|---|---|
| | Obs. | Mod. | MB | R | RMSE | Obs. | Mod. | MB | R | RMSE | Obs. | Mod. | MB | R | RMSE |
| T (°C) | 4.2 | 4.1 | -0.1 | 0.77 | 1.5 | 3.6 | 3.3 | -0.3 | 0.75 | 1.6 | 1.0 | 1.8 | 0.8 | 0.74 | 1.7 |
| WS (m s$^{-1}$) | 1.9 | 2.4 | 0.5 | 0.70 | 1.4 | 1.5 | 2.4 | 0.9 | 0.73 | 1.5 | 1.9 | 2.6 | 0.6 | 0.46 | 1.8 |

Table 2. Performance statistics of PM$_{2.5}$ concentrations.

| Simulated Time Period | Stations | Obs. (µg·m$^{-3}$) | Sim. (µg·m$^{-3}$) | R | MB (µg·m$^{-3}$) | NMB (%) | NME (%) | MFB (%) | MFE (%) |
|---|---|---|---|---|---|---|---|---|---|
| 20:00 Nov. 17– 22, 2012 | CRAES | 121.5 | 190.9 | 0.87 | 69.4 | 57.2 | 185.2 | 53.6 | 60.1 |
| | GY | 139.0 | 289.4 | 0.91 | 150.4 | 108.1 | 183.3 | 65.2 | 68.3 |
| | DL | 101.4 | 112.2 | 0.69 | 10.8 | 10.7 | 85.6 | 15.6 | 39.6 |

Notes: Mean bias: $MB = \frac{1}{n}\sum_{i=1}^{n}(Sim_i - Obs_i)$;

Normalized mean bias: $NMB = \frac{\sum_{i=1}^{N}(Sim_i - Obs_i)}{\sum_{i=1}^{N} Obs_i} \times 100\%$; Normal mean error: $NME = \frac{1}{n}\sum_{i=1}^{n}\frac{|Sim_i - Obs_i|}{Obs_i} \times 100\%$;

Mean fractional bias: $MFB = \frac{1}{N}\sum_{i=1}^{N}\frac{(Sim_i - Obs_i)}{(Obs_i + Sim_i/2)}$; Mean fractional error: $MFE = \frac{1}{N}\sum_{i=1}^{N}\frac{|Sim_i - Obs_i|}{(Obs_i + Sim_i/2)}$.

Table 3. Emission source contribution to the average PM$_{2.5}$ concentration over Beijing on Nov 21.

| Factors | Time period | BJ | BJ-sens | HuaB | HuaB-sens | Emis-intense |
|---|---|---|---|---|---|---|
| SC/PC | d0 | 14.5% | 12.5% | 25.6% | 18.4% | 8.7% |
| | d1 | 20.7% | 18.3% | 57.5% | 37.3% | 19.7% |
| | d2 | 21.3% | 18.8% | 67.1% | 40.9% | 22.4% |

Notes: d0 refers to emission contributions from November 21; d1 refers to emissions contribution from the 20th to the 21st of November; d2 refers to emission contributions from the 19th to the 21st of November.

SC/PC = time cumulative Sensitivity Coefficient/Peak Concentration;

Table 4. Contrast of sensitive (or Emis-intense) and full region emission source contributions.

| | GRAPES-CUACE aerosol adjoint model results | | | | Models-3/CMAQ results (Zhai et al., 2016) | |
|---|---|---|---|---|---|---|
| Time period | Factors | BJ-sens | HuaB-sens | Emis-intense | BJ-sens | HuaB-sens |
| d0 | S/F(effect) | 86.6% | 71.9% | 34.0% | | |
| | S/F(efficiency) | 1.4 | 7.0 | 3.3 | | |
| d1 | S/F(effect) | 88.2% | 64.9% | 34.2% | 99.2% | 93.7% |
| | S/F(efficiency) | 1.5 | 6.3 | 3.3 | 1.8 | 5.3 |
| d2 | S/F(effect) | 88.2% | 61.0% | 33.4% | 100.8% | 87.2% |
| | S/F(efficiency) | 1.5 | 6.0 | 3.3 | 1.9 | 5.0 |

Notes: S/F(effect) = Sensitivity Coefficient over sensitive source region/Sensitivity Coefficient over corresponding full source region;

Contribution Efficiency = Sensitivity Coefficient/Number of region's simulation grid cells;

S/F(efficiency) = Contribution Efficiency of sensitive region/Contribution Efficiency of corresponding full source region.

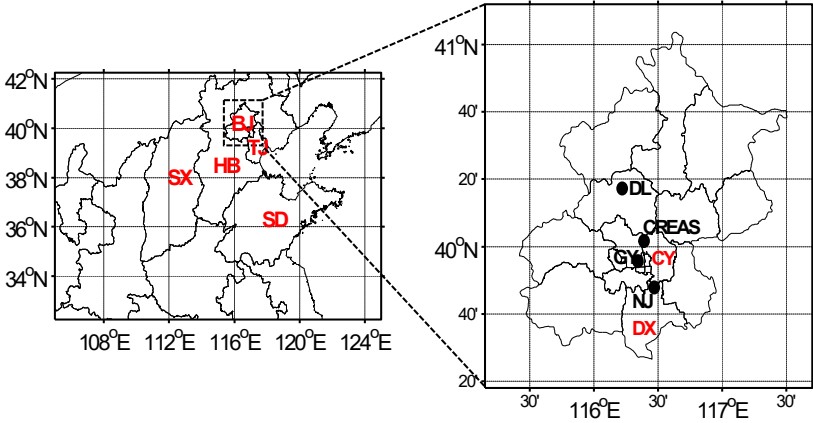

**Figure 1. Left: Model domain and location of Beijing Municipality (BJ), Tianjin Municipality (TJ), Heibei Province (HB), Shandong Province (SD), and Shanxi Province (SX); right: Locations of the Chinese Research Academy of Environmental Sciences (CRAES) station, the Guanyuan (GY) station, the Dingling (DL) station, the Nanjiao (NJ) station, Daxing district (DX) and Chaoyang (CY) district.**

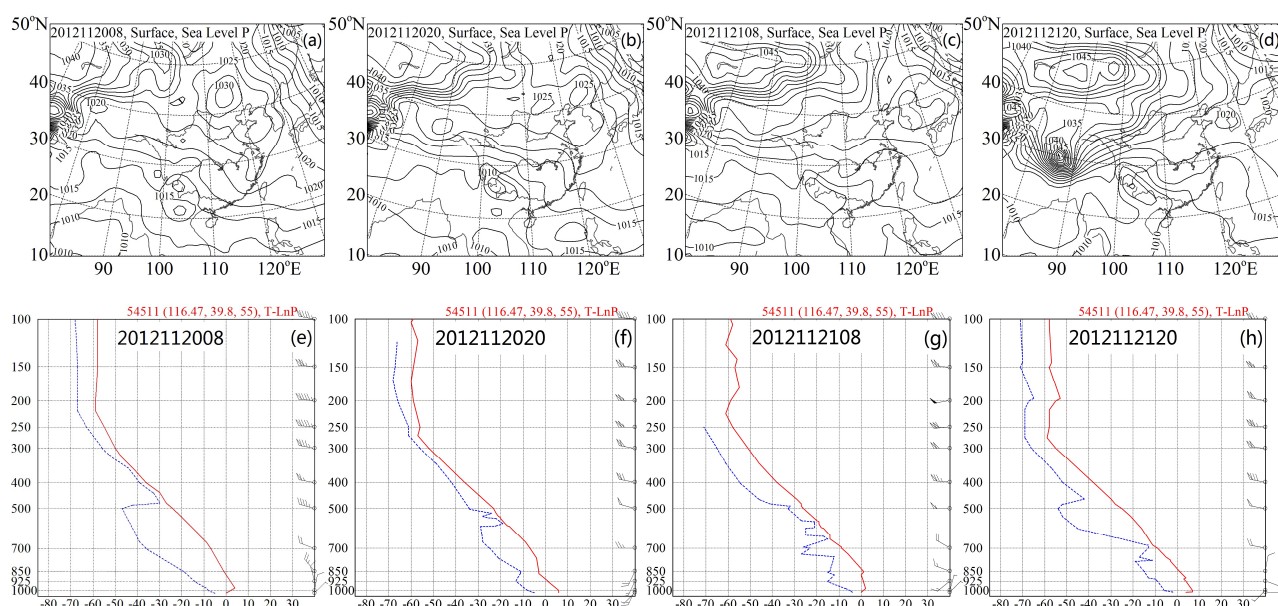

**Figure 2. (a–d): Sea-level pressure field; (e–h): temperature-logarithmic pressure diagrams (blue dotted curves indicate dew point-pressure; red solid curves indicate stratification) at the Nanjiao Station from 08:00 (local time) on November 20, 2012, to 20:00 (local time) on November 21, 2012.**

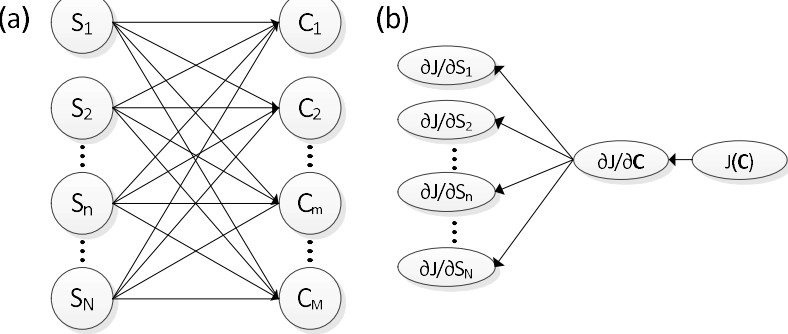

**640**

**Figure 3. Schematic diagrams of the atmospheric chemistry forward (a) and adjoint (b) models. $S_1$, $S_2$, …, $S_n$, …, $S_N$ are emission sources of different sectors, or of different species, at different locations etc., and S is the emission vector; $C_1$, $C_2$, …, $C_m$, …, $C_M$ are pollutant concentrations at different sites, or of different species, and C is the concentration vector.**

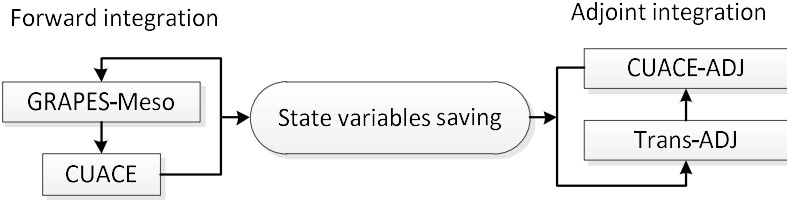

**645**

**Figure 4. Operational processes of the GRAPES-CUACE aerosol adjoint**

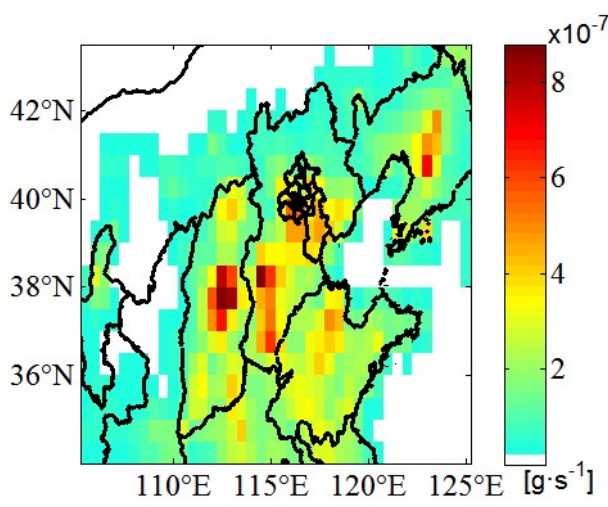

**Figure 5. Gridded distribution of PM$_{2.5}$ primary emission sources.**

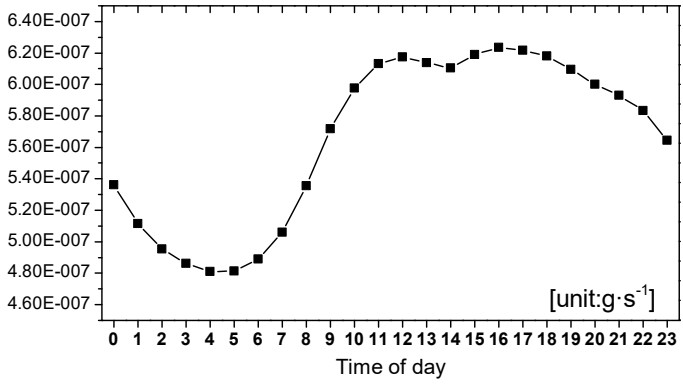

**650**

**Figure 6. Hourly variation in primary PM$_{2.5}$ emission sources in Beijing.**

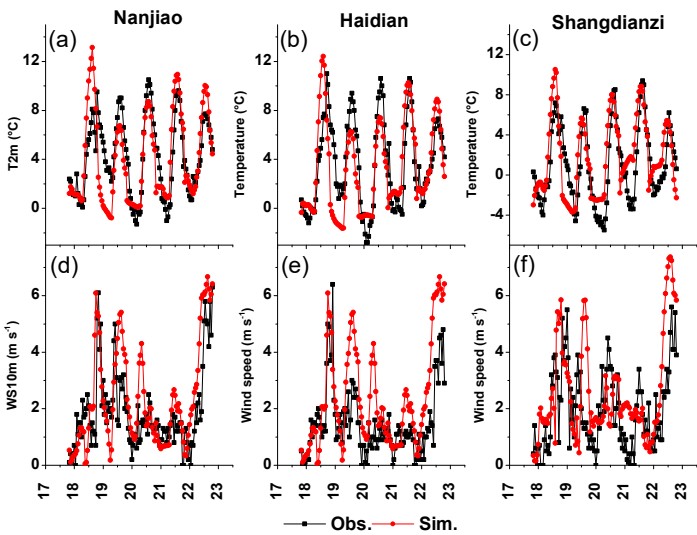

**Figure 7. The temporal variations of observed and simulated hourly 2 m temperature (T2m) (a-c) and 10 m wind speed (WS10m)**
**(e-f) at Nanjiao, Haidian and Shandianzi stations. The observed WS10m are 10-min averaged wind speed.**

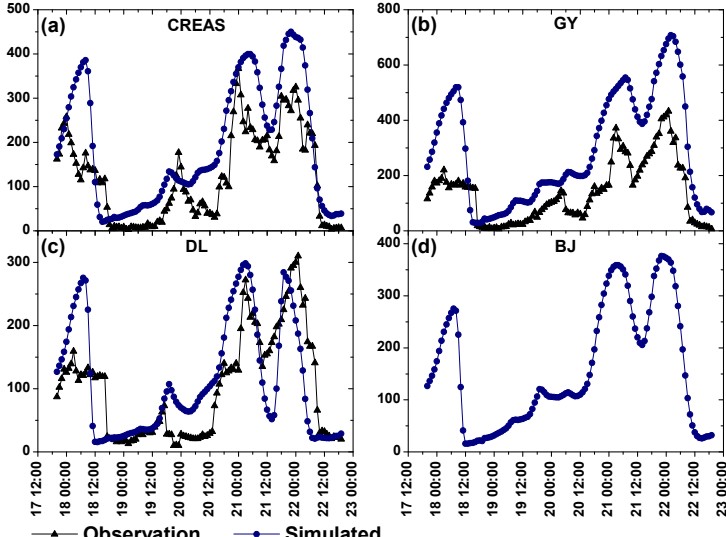

**Figure 8. (a)-(c): Comparisons of the observed (black solid triangles) and simulated (blue dot-line) hourly PM$_{2.5}$ concentrations at the CRAES, GY, and DL stations; (d): Hourly variations in the average PM$_{2.5}$ concentrations over Beijing Municipality.**

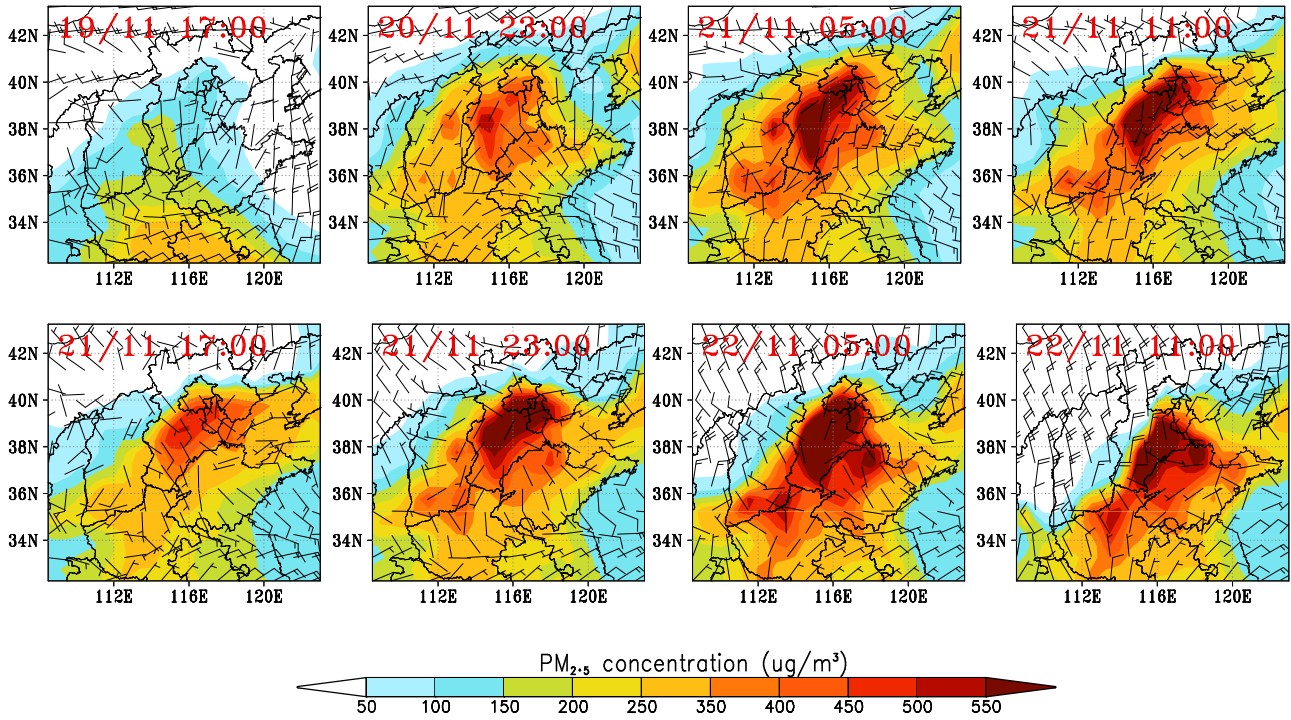

**Figure 9. Variations of simulated surface PM2.5 concentrations and wind field distributions.**

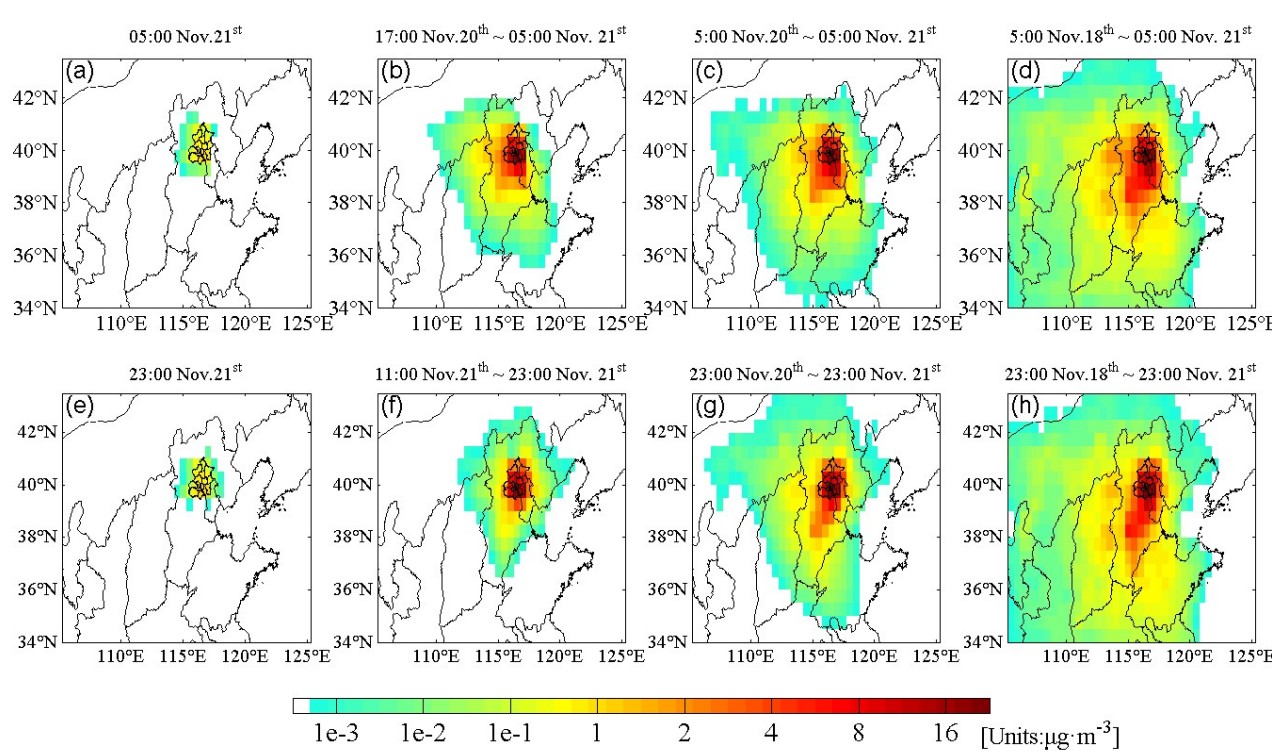

**Figure 10. Time-integrated sensitivity coefficients of surface Beijing PM2.5 concentration peaks to primary PM2.5 sources. (a–d): 1, 12, 24, and 72 h integrated sensitivity coefficients for the 5:00 LT PM2.5 concentration peak on November 21; (e–h): 1, 12, 24, and 72 h integrated sensitivity coefficients for the 23:00 LT PM2.5 concentration peak on November 21.**

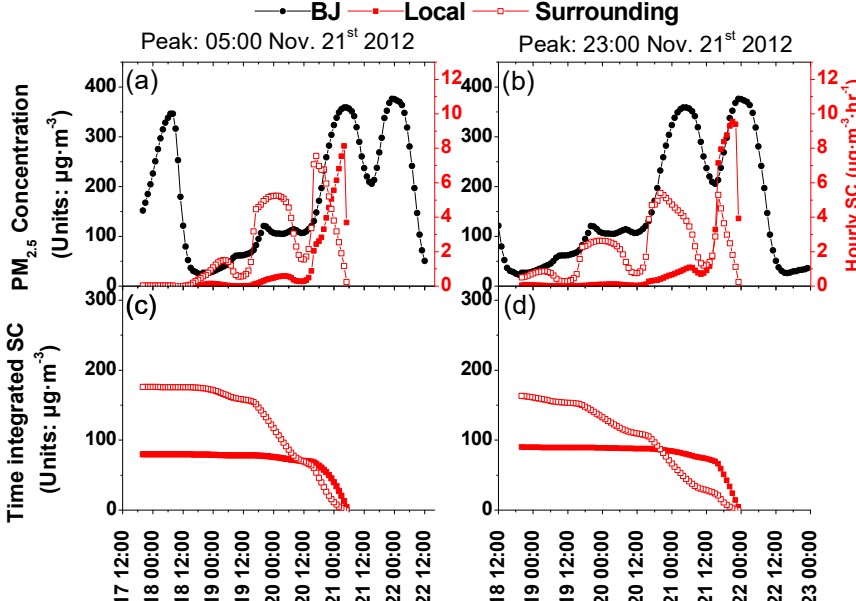

Figure 11. Hourly variations of surface PM$_{2.5}$ concentrations in Beijing and sensitivity coefficients of surface PM$_{2.5}$ concentration peaks in Beijing to local and surrounding primary PM$_{2.5}$ sources. The left and right panels correspond to PM$_{2.5}$ concentration peaks at 05:00 LT and at 23:00 LT on the 21st of November 2012, respectively. (a–b) Hourly variations of Beijing PM$_{2.5}$ concentrations (black solid dot-line) and hourly instantaneous sensitivity coefficients to local (red closed squares) and surrounding (red open squares) emission sources. (c–d) The time-integrated sensitivity coefficients to local (red closed squares) and surrounding (red open squares) emission sources.

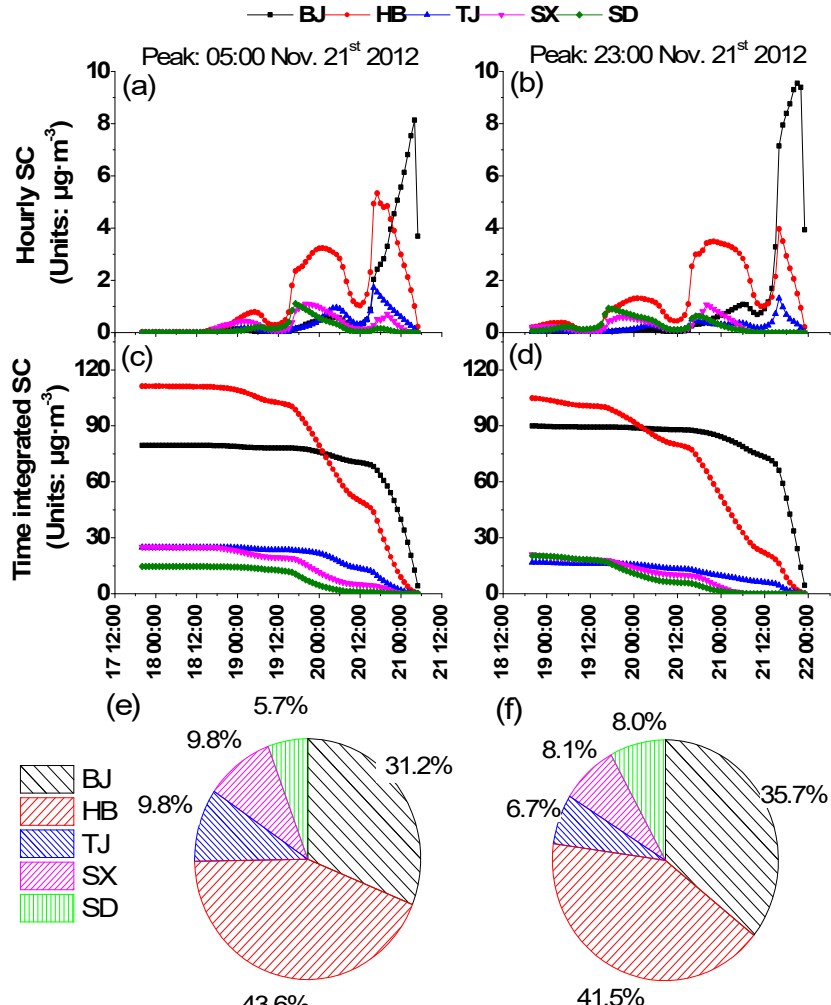

**Figure 12. Sensitivity coefficients of surface PM₂.₅ concentration peaks in Beijing to primary emission sources from local Beijing and each of the surrounding provinces. The left and right panels correspond to PM₂.₅ concentration peaks at 05:00 LT and at 23:00 LT on November 21, 2012, respectively. (a–b) Hourly instantaneous sensitivity coefficients to emission sources from local Beijing, Hebei Province, Tianjin City, Shanxi Province, and Shandong Province. (c–d) The time-integrated sensitivity coefficients to local and surrounding provincial emission sources. (e–f) The contribution ratios of emission sources from each surrounding province to PM₂.₅ concentration peaks.**

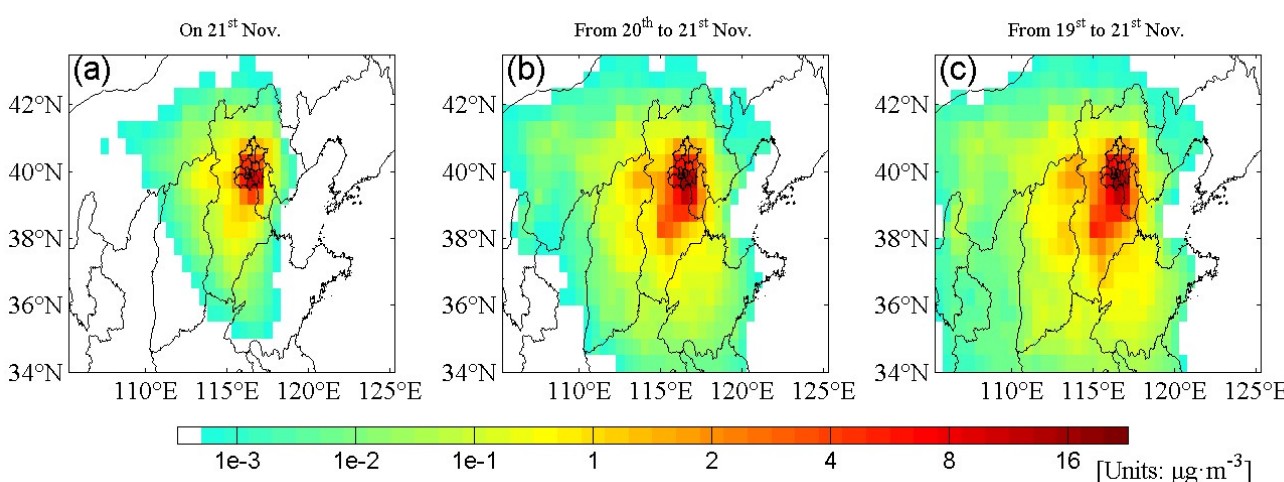

**Figure 13. The 24 (a), 48 (b), and 72 h (c) integrated sensitivity coefficients of surface PM₂.₅ concentrations to primary emission sources in Beijing on November 21, 2012.**

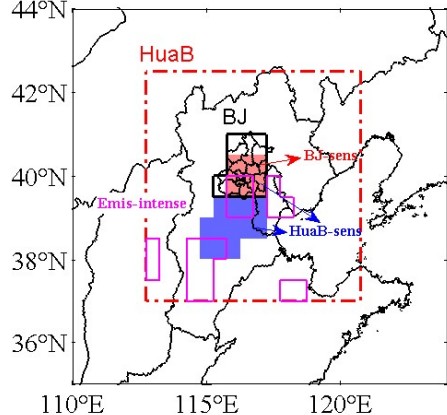

| Regions | Number of grid cells | Sensitive area ratios (%) |
|---|---|---|
| HuaB-sens | 18 | 10.2 |
| HuaB | 176 | |
| BJ-sens | 6 | 60.0 |
| BJ | 10 | |
| Emis-intense | 18 | 10.2 |

**Figure 14. Domain definition of Huabei (HuaB, in red dot-dashed frame), Beijing (BJ, in black solid frame), sensitive Beijing (BJ-sens, red shaded), sensitive Huabei (HuaB-sens, both red and blue shaded), and emission intensive (Emis-intense, in pink solid frame) regions.**

**Notes: HuaB-sens area ratio = HuaB-sens floor space/HuaB floor space × 100%;**

**BJ-sens area ratio = BJ-sens floor space/BJ floor space × 100%;**

**Emis-intense area ratio = Emis-intense floor space/HuaB floor space × 100%.**