# Peer review of "Detection of critical PM2.5 emission sources and their contributions to a heavy haze episode in Beijing, China, using an adjoint model"

_Atmospheric Chemistry and Physics, 2016_

## Referee Comment (RC1) · Anonymous Referee #2 · 13 Jan 2017

General comments

The authors attempted to determine the contributions from local and surrounding emission to two PM2.5 peaks during a heavy Beijing haze episode by using an aerosol adjoint model. Sensitivity analysis of the model simulations was performed to detect the PM concentration-source relationship by examining the temporal variation of an user-defined sensitivity coefficient and its time-integrated values. Given that there are still debates on the relative contributions of aerosols from local emission and regional transport to Beijing haze, the adjoint modeling studies and sensitivity analysis in this study would be interesting to the readerships of the ACP journal. However, some issues related to the clarity of discussions need to be addressed before its publication.

[Figure]

Specific comments

1) According to Fig. 4, what the reasons for the significant decrease of PM concentration during 11:00 to 17:00 on Nov. 21st? Could it be the development of PBL or the reduction of emissions during this period? 2) On p. 10 lines 2-5, the authors attributed the overall higher contribution from the surrounding emissions than local emissions to the obvious periodic fluctuation of hourly sensitivity coefficient of surrounding emissions. This explanation is not convincible for me since we only can infer that there was larger temporal variation for the contribution of surrounding emissions than that of local emissions based on the fluctuation of sensitivity coefficient. 3) I would suggest the authors to move the detailed discussions about computational efficiency of the adjoint model and the Models-3/CMAQ systems in the conclusion section (p. 12 lines 3-9) to section 4.3. Just a brief and concise summary is needed for the model computational efficiency in the conclusion section. 4) It was stated that the threshold to determine sensitive emission regions was based on the relative magnitude of sensitive coefficients and the sources contribution ratios of sensitive regions to the objective function (p. 10 lines 21-23). What are the exact values for the relative magnitude of sensitive coefficients and the sources contribution ratios of sensitive regions to the objective function? Otherwise, I feel that the selection of the threshold is arbitrary.

Technical corrections

1) On p. 11 line 19: remove the first 'peak'. 2) The text in Fig. 1 was not legible. Please enlarge the font size. 3) For Fig. 6, the Y axis label of "PM2.5 concentration" overlays on the one for panel (c). Adjust its position to where is only for panels (a) and (b). 4) P. 6 line 8: 'might attribute' should be 'might be attributed'. 5) P. 6 line 10: I would like to use 'have proven' to replace 'had convinced'. 6) Please pay attention to the tense consistency through the manuscript. For example, on p. 6 line 28, it used both current and pass tenses (note the words 'take' and 'defined').

---

## Short Comment (SC1) · 13 Feb 2017

As for the regional and local contributions for the Beijing haze pollution, we also launched some observation studies, but were not verified by model simulations [G. Tang et al., 2015; Zhu et al., 2016]. After I read this paper, I was so glad that the simulated results were consistent with our findings. However, there were some tiny questions that I want to discuss with the authors:

a. The authors investigated the haze pollution in Beijing based on the two peaks of PM2.5 concentrations, and the PM2.5 concentrations were peaked on 05:00 BT and 23:00 BT NOV. 21st 2012 respectively. However, these two moments are probably the high pollutant concentration periods originally due to some local emissions (such as

the diesel cars). Does it necessary to divide the period so detail? I suppose if it was better to divide the haze episode into two stages, the first stage (from NOV. 19 to 20 2012) may be defined as the early polluted period and the second stage (included the two peaks) may be defined as the heavy polluted period. And then to quantify the regional transport and local contributions.

b. Does the obvious periodic fluctuation of hourly sensitivity coefficient of surrounding emissions has some relationship with the mountain-valley winds [Guiqian Tang et al., 2016]?

Tang, G., et al. (2016), Mixing layer height and its implications for air pollution over Beijing, China, Atmospheric Chemistry and Physics, 16(4), 2459-2475, doi:10.5194/acp-16-2459-2016.

Tang, G., X. Zhu, B. Hu, J. Xin, L. Wang, C. Münkel, G. Mao, and Y. Wang (2015), Impact of emission controls on air quality in Beijing during APEC 2014: lidar ceilometer observations, Atmospheric Chemistry and Physics, 15(21), 12667-12680, doi:10.5194/acp-15-12667-2015.

Zhu, X., G. Tang, B. Hu, L. Wang, J. Xin, J. Zhang, Z. Liu, C. Münkel, and Y. Wang (2016), Regional pollution and its formation mechanism over North China Plain: A case study with ceilometer observations and model simulations, Journal of Geophysical Research: Atmospheres, 2016JD025730, doi:10.1002/2016JD025730.

---

## Author Comment (AC1) · 14 Feb 2017

Dear referee,

Thank you very much for your valuable comments. Revisions made in the manuscript are in red characters or highlighted. Responses to the comments are listed as followings:

General comments

The authors attempted to determine the contributions from local and surrounding emission to two PM2.5 peaks during a heavy Beijing haze episode by using an aerosol adjoint model. Sensitivity analysis of the model simulations was performed to detect the

[Figure]

PM concentration-source relationship by examining the temporal variation of a user-defined sensitivity coefficient and its time-integrated values. Given that there are still debates on the relative contributions of aerosols from local emission and regional transport to Beijing haze, the adjoint modeling studies and sensitivity analysis in this study would be interesting to the readerships of the ACP journal. However, some issues related to the clarity of discussions need to be addressed before its publication.

Specific comments

1) According to Fig. 4, what the reasons for the significant decrease of PM concentration during 11:00 to 17:00 on Nov. 21st? Could it be the development of PBL or the reduction of emissions during this period?

Response: Decrease of PM2.5 concentration from morning to afternoon is typical in Beijing, resulted mainly from diurnal variation of (planetary boundary layer) PBL with the development of vertical mixing after sunrise for diluting pollutants (Zhao et al., 2009; Liu et al., 2015). Meanwhile, the emissions could be reduced during this period (An et al., 2013).

P6: lines 20-26 in section 3.4 were revised as: Thereafter, PM2.5 concentration over Beijing, south central Hebei and Tianjin decreases to a trough in the Nov. 21st afternoon before rising above $550\mu g/m3$ at 23:00 BT. Decrease of PM2.5 concentration from morning to afternoon is typical in Beijing, resulted mainly from diurnal variation of (planetary boundary layer) PBL with the development of vertical mixing after sunrise for diluting pollutants (Zhao et al., 2009; Liu et al., 2015). Meanwhile, the emissions could be reduced during this period (An et al., 2013). The concentration peak at 23:00 BT owns to the influence of the easterly winds on Nov. 21st, which causes pollutants convergence ahead of the Tai-hang Mountains and carries abundant water vapor that promotes local hygroscopic growth.

References:

An, X. Q., Sun, Z. B., Lin, W. L., Jin, M., and Li, N.: Emission inventory evaluation using observations of regional atmospheric background stations of China, J. Environ. Sci., 25, 537-546, 2013.

Added references:

Liu, Z., Hu, B., Wang, L., Wu, F., Gao, W., and Wang, Y.: Seasonal and diurnal variation in particulate matter (PM10 and PM2.5) at an urban site of Beijing: analyses from a 9-year study, Environmental Science and Pollution Research, 22, 627-642, 2015.

Zhao, X., Zhang, X., Xu, X., Xu, J., Meng, W., and Pu, W.: Seasonal and diurnal variations of ambient PM2.5 concentration in urban and rural environments in Beijing, Atmospheric Environment, 43, 2893-2900, 2009.

2) On p. 10 lines 2-5, the authors attributed the overall higher contribution from the surrounding emissions than local emissions to the obvious periodic fluctuation of hourly sensitivity coefficient of surrounding emissions. This explanation is not convincible for me since we only can infer that there was larger temporal variation for the contribution of surrounding emissions than that of local emissions based on the fluctuation of sensitivity coefficient.

Response: Thank you for the comments. Here we are aiming to find out the more influential regions and time period of emissions from the hourly sensitivity coefficients. Since we can infer 'that there is larger temporal variation for the contribution of surrounding emissions than that of local emissions based on the fluctuation of sensitivity coefficient', surrounding emissions emitted 2-3 days ahead of the pollution peak is of great importance to the pollution peak . In this way, joint control of emissions with surrounding areas ahead of the polluted peak can lead to effective result.

To clarify this statement, we have revised these lines to: From the time series of hourly sensitivity coefficients for local and surrounding emissions (Fig. 6a and b), we can see that the temporal variation in the contribution of surrounding emissions is larger than

that of local emissions during 2-3 days ahead of the pollution peak, which indicates a notable contribution of surrounding emissions to PM2.5 peak during this time period. Therefore, if joint control of pollutant emissions with Hebei province, Tianjin city and Shanxi province were implemented 2-3 days ahead of the first PM2.5 concentration peak, then PM2.5 concentration accumulation due to transported pollutants could be effectively prevented, thus decreasing the concentration of these two PM2.5 concentration peaks.

3) I would suggest the authors to move the detailed discussions about computational efficiency of the adjoint model and the Models-3/CMAQ systems in the conclusion section (p. 12 lines 3-9) to section 4.3. Just a brief and concise summary is needed for the model computational efficiency in the conclusion section.

Response: Thank you for your valuable suggestion, this will definitely make section 4.3 more complete and the conclusion part concise.

We have moved the following content as an added paragraph at the end of section 4.3: Beyond that, the computational loads of the adjoint simulation are much smaller than the assessments with Models-3/CMAQ modeling (Zhai et al., 2016). For the adjoint simulation, one forward integration (for un-equilibrated data saving) and one backward adjoint integration can obtain the influence of emissions from any source region, during any time period to PM2.5 peaks. However, in the Models-3/CMAQ assessments, in order to compare the effects of emission reductions over two different time periods, at two different ratios and over four different regions, 12 sensitivity tests are set and the forward model is integrated for 13 times (one control simulation included).

We also replaced the above contents in the conclusion with a concise sentence: Meanwhile, the adjoint simulation is of much higher computational efficiency than the assessments with Models-3/CMAQ modeling.

4) It was stated that the threshold to determine sensitive emission regions was based on the relative magnitude of sensitive coefficients and the sources contribution ratios

of sensitive regions to the objective function (p. 10 lines 21-23). What are the exact values for the relative magnitude of sensitive coefficients and the sources contribution ratios of sensitive regions to the objective function? Otherwise, I feel that the selection of the threshold is arbitrary.

Response: Thank you very much for pointing this out. Here, we want to express the threshold in order to compare the adjoint results and the Models-3/CMAQ assessments (Zhai et al., 2016), the full administrative regions (HuaB and BJ) and the sensitive regions (HuaB-sens and BJ-sens) are selected based on the regions definition in the previous research (Zhai et al., 2016) and the adjoint results. The reason why we choose $3\mu g/m3$ as the threshold is that in this way, the HuaB-sens accounts for 10.2% the area of HuaB and the BJ-sens accounts for 60.0% the area of BJ, similar with the sensitive and administrative regions definition by Zhai et al. (2016), thus making the results comparable.

We have revised the the corresponding description to: In order to compare the adjoint results with the Models-3/CMAQ assessments, we refer to the research by Zhai et al. (2016) and select four emission regions based on administrative divisions and the adjoint results: the overall Huabei region (HuaB), the sensitive Huabei region (HuaB-sens), the overall Beijing municipality (BJ), and the sensitive Beijing region (BJ-sens) (Fig. 8). In this research, grid cells with 72-h cumulated sensitivity coefficient larger than $3\mu g/m3$ are covered by the sensitive emission regions (HuaB-sens and BJ-sens), and grid cells with smaller sensitive values are outside the sensitive emission regions. This means that emissions within the sensitive emission regions have relatively larger impact on the PM2.5 concentration peak. Here, the HuaB-sens accounts for 10.2% the area of HuaB and the BJ-sens accounts for 60.0% the area of BJ (Fig. 8), similar with the sensitive and administrative regions definition in research by Zhai et al. (2016).

Technical corrections

1) On p. 11 line 19: remove the first 'peak'.

[Figure]

Response: The 'peak' is removed.

2) The text in Fig. 1 was not legible. Please enlarge the font size.

Response: The brief explanations of Fig. 1 were added in the Fig. 1 caption as follows: Figure 1. (a-d): Sea-level pressure field (black contour lines; Beijing is marked with a red triangle); (e-h): temperature-logarithmic pressure diagrams (thick red solid curves for: process; green solid curves for: dew point-pressure; blue solid curve for: stratification) at Nanjiao Station from 08:00 (local time) on Nov. 20th 2012 to 20:00 (local time) on Nov. 21st 2012. Detailed information of Fig. 1 is found in Fig. S1 in the supplement. Meanwhile, enlarged figures and explanations were added in Fig. S1 in the supplement. Correspondingly, 'Fig. S1' is revised to 'Fig. S2', and 'Fig. S2' is revised to 'Fig. S3' in the manuscript.

3) For Fig. 6, the Y axis label of "PM2.5 concentration" overlays on the one for panel (c). Adjust its position to where is only for panels (a) and (b).

Response: The position of the Y axis label of "PM2.5 concentration" was adjusted.

4) P. 6 line 8: 'might attribute' should be 'might be attributed'.

Response: 'might attribute' was changed to 'might be attributed'.

5) P. 6 line 10: I would like to use 'have proven' to replace 'had convinced'.

Response: Thank you for your suggestion. The 'had convinced' was replaced by 'have proven'.

6) Please pay attention to the tense consistency through the manuscript. For example, on p. 6 line 28, it used both current and pass tenses (note the words 'take' and 'defined').

Response: We have revised the whole manuscript to current tense. Revised words are in red color.

[Figure]

**[ACPD]{.acpd}**

Interactive
comment

Other minor revisions

P2 line 6: 'to any model parameters' is changed to 'to model parameters'.

P3 line 27: 'the ahead 2 days' is changed to 'the previous 2 days'.

P4 line 27: 'reflect exact contribution' is revised to 'reflect the absolute contribution'.

Please also note the supplement to this comment:
http://www.atmos-chem-phys-discuss.net/acp-2016-911/acp-2016-911-AC1-supplement.zip

---

## Referee Comment (RC2) · Anonymous Referee #3 · 21 Feb 2017

**1   Overview**

The manuscript by Zhai et al. investigates sources of PM$_{2.5}$ for a pollution episode in Beijing using adjoint modeling. The work is a nice start, and a good use of the new tools that this group has developed. However, the manuscript overall feels a bit premature; it reads like a first draft. The overall purpose of using the adjoint model is not well articulated, nor is the tool used to its full potential. The comparisons and evaluations to observations and other studies are often qualitative and not particularly well fleshed out, and the presentation of results is murky in a few critical areas. The manuscript also requires substantial grammatical editing throughout. It is possible the

work would be suitable for ACP after major revisions, but a different journal such as Atmospheric Research may be a better fit.

**2 Major comments**

- 2.10-14: While it is true that an adjoint model provides more precise estimates of the sensitivity (partial derivative), this in some cases may also be viewed as a downside compared to perturbation approaches when performing sensitivity calculations for the purpose of source attribution, since the adjoint model fails to capture the nonlinear response of atmospheric chemistry to substantial changes in emissions. Overall, the topic of how these types of sensitivities are interpreted for source contributions needs to be directly addressed in the introduction and methods, and expanded upon in the interpretation of results in more detail.

- Introduction: Several previous studies of source contributions to $PM_{2.5}$ in Beijing are mentioned, but they are only discussed in terms of their computational methods. That would be fine if this paper was in G.M.D. and strictly a discussion of methods. But for a scientific paper in ACP, the authors need to discuss the actual scientific findings of previous works. They need to clearly articulate what has previously been written about the sources that contribute to Beijing $PM_{2.5}$, and how their current study will advance the understanding of sources (most likely by providing insights into the spatial variability of contributions that can be most readily obtained using adjoint methods). Some justification for studying the specific pollution episode of Nov 19-21, 2012, also needs to be provided.

- 5.26: The authors claim that the initial concentrations and boundary conditions are set as the "observed monthly means", but this does not make sense, as it is impossible that the concentrations of all species were observed at all locations throughout the domain in order to established an observationally-derived initial

condition and boundary condition. Thus, please describe in more detail how initial and boundary conditions are estimated.

- Section 3: Please include an entire new section covering in detail the emissions (anthropogenic and natural) used in this model, including a description of their daily and hourly variability. These are critical for understanding the significance of sensitivities of the form $\frac{\partial J}{\partial S_n} S_n$.

- Fig 3(a) / Section 3.3: The discussion model performance evaluation needs to be improved and expanded. It appears that the simulation over-estimates $PM_{2.5}$ concentrations are overestimated, although the timing of the peaks is well-correlated with the measurements. Are there no measurements on Beijing site to compare with? Are only measurements of total $PM_{2.5}$ available? How well does this model do at reproducing concentrations of specific aerosol components, such as BC, sulfate, nitrate, etc.? If this has been documented in previous work for Beijing specifically, then the authors should be more quantitative when discussing the model skill using metrics such as normalized mean bias, normalized mean error, etc. It is also interesting that the model over-estimates measurements, given that many air quality models fail to represent the high levels of $PM_{2.5}$ concentrations observed during peak episodes in Beijing owing to missing treatment of heterogenous chemistry, as described in several recent papers such as Wang et al. (PNAS, 2016, doi:10.1073/pnas.1616540113) and Cheng et al. (Science Advances, 2016, doi:10.1126/sciadv.1601530).

- Section 4.1: There are several species and sectors that have emissions that contribute to $PM_{2.5}$ formation. Which emissions are considered in the presentation of the results here? In other works, how is $S_n$ defined? Are anthropogenic and natural sources included? What type of anthropogenic sources? Is it the total emission across all species? This is an essential missing detail. The results have little scientific or policy relevance in current form, given that they are only

presented in terms of local vs nonlocal sources (a point for which use of an adjoint model would be overkill).

- 8.9-11: This statement hasn't really been demonstrated. To use the adjoint sensitivities to "reproduce" the air pollution episode, one would need multiply the time series of sensitivities by the time series of emissions and show that their product matches the observations. This has not been done, nor would it likely work owing to nonlinearities. Claims of efficiency are also implied but not quantified. A single adjoint model integration is often several times (2 - 10) slower than a normal forward model integration. Thus what is the overall computational savings of their approach here over forwarfd methods, given the size of $N$ and $M$, quantitatively?

- Fig 6(e) and (f) are good to know, but they are somewhat of a waste of an adjoint model. If the only interest was in the separation between "surrounding" vs "local" emissions of all PM$_{2.5}$ precursor emissions, this could have been achieved with only 3 forward model integrations (adjoint not needed). So the authors haven't really brought out the strength of their results to provide insight into spatial attributions beyond these two regions. Pie chart showing the influence by province, species, and sector would be much more interesting, and would start to approach a level of detail unobtainable without use of an adjoint model.

- Fig 8: Defining these ratios based on the area of the regions is not the best idea. It would be better to define the ratios based on the magnitude of the emissions in the different regions, since emissions intensity per unit area is not uniform.

- 10.25-28: Table 1 and the argument based on area isn't a great method, as discussed above. And I'm sorry but Table 2 and surrounding discussion just does not make much sense, and requires further clear explanation of what is being presented. What is the importance of the ration SC / PC? This needs to be explained. What are the percent values percentages of? Do these sum to 100% in some manner? Lastly, comparison to results of Zhai et al. (2016) appears to be

entirely qualitative, and no clear summary of how the two compare quantitatively is provided.

**3 Minor comments**

- 2.2: The first sentence is a bit vague, and should be clarified. Adjoint models are efficient for some types of sensitivity calculations, but not all. They are also efficient in terms of wall-time, but not necessarily in terms of memory or i/o.

- 2.14-19: This brief overview of "current" applications of adjoint modeling in atmospheric chemistry isn't a great fit for this paper, as it doesn't cover the first works in this area, historically, nor is it limited to only the latest works. Also, in attempting to cover all applications of adjoint model, the authors touch upon several areas (O3, CO, etc.) that aren't directly related to the topic of PM$_{2.5}$. I suggest the authors instead consider a more detailed overview of previous works, but one that is more narrowly limited in terms of scope, possibly to just sensitivity studies of PM$_{2.5}$.

- 3.2: There is a second paper by the same group using adjoint modeling to investigate sources of PM$_{2.5}$ in Beijing during the APEC period.

- 3.9: Could the authors clarify what is meant by "guidance on the enaction of dynamic environmental control policy"? What type of policy are they referring to (municipal? national? international?), and what is dynamic about such policy?

- 4.13: Technically a first-order finite difference calculation would require $N$+1 forward model integrations.

- 4.18: The theoretical equivalence of these approaches predates the work of Liu by many decades; I suggest the authors find a more fundamental reference. Also,

it is typical to only cite PhD thesis (as opposed to peer reviewed literature) when absolutely necessary, which is not the case here.

- 4.26: Adjoint sensitivities would only provide "exact" contributions for linear systems. However, $PM_{2.5}$ is formed nonlinearly, which needs to be addressed, or the interpretation and use of the adjoint sensitivities needs to be reconsidered.

- 5.14: "Unequilibrated" is not the correct word here. Nonlinear?

- Section 3.2: In addition to the physical processes treated in this aerosol model, please also briefly review what chemistry is included, both in the aerosol and gas-phase, and how the thermodynamic partitioning of species across phases is modeled.

- Section 3.3: Previous studies have shown that there are influences of emissions on $PM_{2.5}$ in your receptor cite from beyond the model domain considered here. Thus please explain how the influence from boundary conditions is tracked in the adjoint modeling.

- 7.12: This is a more correct interpretation of adjoint sensitivity results which should be considered in the earlier descriptions.

- Fig 5: It appears the emissions continue to spread by 72 hours of back integration. How then did the authors decide to stop the adjoint integration at 72 hrs? In other words, why did they not integrate backwards further in time? The lifetime of aerosols can be much longer than 3 days, so integration of back to a week to 10 days may be necessary to capture all non-local influences.

- Section 4.1: Please clearly define what is meant by "local" in this context. Is it just the single grid cell that contains the Beijing receptor cite?

- 9.24-26: The non-local contributions do get small after 72 hrs, but as shown in Fig 6(d), the cumulative sensitivities have yet to asymptote to a constant value, which would indicate that sensitivities from early than 72 hrs may still play some role, although small. Also, sensitivities may have transferred to the boundary conditions, as mentioned previously.

**4  Corrections**

I started making grammatical corrections to the abstract, but stopped after only a few lines, as the entire manuscript needs substantial editing.

- 1.17: in detecting → to detect

- 1.20: south to → south of

- 1.21: at the south to → to the south of

---

## Author Response (AR1)

Responses to referee #3 and short comment #1 on "Detection of critical PM$_{2.5}$ emission sources and their contributions to a heavy haze episode in Beijing, China, using an adjoint model" **and marked-up version of the manuscript.**

**To Referee #3**

Dear referee,
Thank you very much for your valuable comments. Revisions in the track change version of the manuscript are in red or shaded. This document is organized as follows: the referee's comments are in black and responses to the comments are in blue.

**1. Overview**

The manuscript by Zhai et al. investigates sources of PM$_{2.5}$ for a pollution episode in Beijing using adjoint modeling. The work is a nice start, and a good use of the new tools that this group has developed. However, the manuscript overall feels a bit premature; it reads like a first draft. The overall purpose of using the adjoint model is not well articulated, nor is the tool used to its full potential. The comparisons and evaluations to observations and other studies are often qualitative and not particularly well fleshed out, and the presentation of results is murky in a few critical areas. The manuscript also requires substantial grammatical editing throughout. It is possible the work would be suitable for ACP after major revisions, but a different journal such as Atmospheric Research may be a better fit.

Response: Thank you very much for your comments. This study selected the same pollution episode as that in a previous work (Zhai et al., 2016) carried out by our group. In contrast to the previous work, the overall purpose of using the **backward** adjoint model is to provide detailed temporal-spatial variation of PM$_{2.5}$ critical sources with high computational efficiency, which is almost infeasible to be obtained through the **forward** emissions off-and-on sensitivity analysis (the perturbation approach) due to high computational cost. This overall purpose is then added in the **introduction**. To take better advantage of the adjoint model, we then divided emission sources impacts on Beijing PM$_{2.5}$ into different provinces around Beijing (added in section 4.4). In addition, model evaluations are strengthened in section 3.3 with measurements from two more observational sites (one rural station and one urban station) and qualitative statistical metrics of model performance are added. The whole manuscript is edited by the Enago (http://www.enago.com) English editing services and verified by the authors. The authors will also order the English copy-editing services from Copernicus.

**2. Major comments**

• 2.10-14: While it is true that an adjoint model provides more precise estimates of the sensitivity (partial derivative), this in some cases may also be viewed as a downside compared to perturbation approaches when performing sensitivity calculations for the purpose of source attribution, since the adjoint model fails to capture the nonlinear response of atmospheric chemistry to **substantial changes** in emissions. Overall, the topic of how these types of sensitivities are interpreted for source contributions needs to be directly addressed in the introduction and methods, and expanded upon in the interpretation of results in more detail.

Response: Thank you very much for your valuable comments. We have made the corresponding revisions in the introduction, methods and interpretation of results. The revised presentation **emphasized** that the adjoint sensitivity reveals the changes in concentration due to small perturbations in emissions sources, in other words, the adjoint model estimates the incremental influence of specific sources on air quality attainment. Corresponding revisions are green shaded.

• Introduction: Several previous studies of source contributions to PM$_{2.5}$ in Beijing are mentioned, but

they are only discussed in terms of their computational methods. That would be fine if this paper was in G.M.D. and strictly a discussion of methods. But for a scientific paper in ACP, the authors need to discuss the actual scientific findings of previous works. They need to clearly articulate what has previously been written about the sources that contribute to Beijing PM$_{2.5}$, and how their current study will advance the understanding of sources (most likely by providing insights into the spatial variability of contributions that can be most readily obtained using adjoint methods). Some justification for studying the specific pollution episode of Nov 19-21, 2012, also needs to be provided.

Response: Following the referee's comments, we have deleted the paragraph that describes current applications of adjoint modeling in atmospheric chemistry and added new information to discuss the actual scientific findings of previous works regarding sources that contribute to Beijing PM$_{2.5}$. We then illustrated previous studies that implement the adjoint model in Beijing air pollutants tracking, which proved the high efficiency and accuracy of an atmospheric chemistry adjoint model in Beijing air pollutants source apportionment. Thereafter, justification for studying the Nov 19-21, 2012 pollution episode is provided. In the end, we point out that our work advances the understanding of Beijing PM$_{2.5}$ sources by providing insights into the spatial and temporal variability of emission source contributions from each of the surrounding provinces as well as from local and environs transports. Revisions are in red or shaded in the manuscript.

• 5.26: The authors claim that the initial concentrations and boundary conditions are set as the "observed monthly means", but this does not make sense, as it is impossible that the concentrations of all species were observed at all locations throughout the domain in order to established an observationally-derived initial condition and boundary condition. Thus, please describe in more detail how initial and boundary conditions are estimated.

Response: Thank you very much for pointing this incorrect presentation out. Here the initial and boundary values for O$_3$ and OH radical were taken from climatic means, and zeros for each aerosol species (Zhou et al., 2012). Revisions in section 3.3 are in red in the first paragraph.

Hope it is reasonable now.

• Section 3: Please include an entire new section covering in detail the emissions (anthropogenic and natural) used in this model, including a description of their daily and hourly variability. These are

critical for understanding the significance of sensitivies of the form $\frac{\partial J}{\partial s_n} S_n$.

Response: Thank you very much for your comments. Detailed description of emissions used in this model is added in the second paragraph of section 3.3 (red and underlined in the manuscript). Gridded distribution of overall PM$_{2.5}$ primary emission sources is illustrated in Fig. 2, and the hourly variability of the overall particle sources (as well as sulphate as an example) in Beijing is shown in Fig. 3. Description of Fig. 2 and Fig. 3 are added in the third paragraph in section 3.3 (red and underlined in the manuscript).

[Figure]

Figure 2. Gridded distribution of PM$_{2.5}$ primary emission sources.

[Figure]

Figure 3. Hourly variation of primary PM$_{2.5}$ emission sources in Beijing.

• Fig 3(a) Section 3.3: The discussion model performance evaluation needs to be improved and expanded. It appears that the simulation over-estimates PM$_{2.5}$ concentrations, although the timing of the peaks is well-correlated with the measurements. Are there no measurements on Beijing site to compare with? Are only measurements of total PM$_{2.5}$ available? How well does this model do at reproducing concentrations of specific aerosol components, such as BC, sulfate, nitrate, etc.? If this has been documented in previous work for Beijing specifically, then the authors should be more quantitative when discussing the model skill using metrics such as normalized mean bias, normalized mean error, etc. It is also interesting that the model over-estimates measurements, given that many air quality models fail to represent the high levels of PM$_{2.5}$ concentrations observed during peak episodes in Beijing owing to missing treatment of heterogenous chemistry, as described in several recent papers such as Wang et al. (PNAS, 2016, doi:10.1073/pnas.1616540113) and Cheng et al. (Science Advances, 2016, doi:10.1126/sciadv.1601530).

Response: The hourly PM$_{2.5}$ concentration measurements at Guanyuan (GY), an urban site, and Dingling (DL), a rural observation site, are added to evaluate the model performance. Meanwhile, the performance statistics including correlation coefficient (R), mean bias (MB), normalized mean bias (NMB), normalized mean error (NME), mean fractional bias (MFB) and mean fractional error (MFE) are also listed in Table 1 and accordingly analyzed in the context in red. We didn't obtain the available measurements for specific aerosol components. The GRAPES-CUACE modeling system is an on-line coupled meteorology-chemistry modeling system which adopts the size-segregated multicomponent aerosol algorithm in its aerosol module. Except for ammonium, sulphates, BC, OC, sand/dust, nitrates and sea salts are segregated into 12 size bins. Therefore, it has better performance in simulating aerosol concentrations.

Revisions in the manuscript:

① Model performance discussions are in red in section 3.3.

② Locations of GY and DL stations are added in Figure 4:

[Figure]

Figure 4. Left: Model domain settings and location of Beijing municipality (BJ), Tianjin municipality (TJ), Heibei province (HB), Shandong province (SD) and Shanxi province (SX); right: Locations of the Chinese Research Academy of Environmental Sciences (CREAS) station, the Guanyuan (GY) station, the Dingling (DL) station, the Nanjiao (NJ) station, Daxing district (DX) and Chaoyang (CY) district.

③ Hourly concentration curves at GY and DL stations are added in Figure 5b and 5c:

[Figure]

Figure 5. (a)-(c): Comparisons of the observed (black solid triangles) and simulated (blue dot-line) hourly PM$_{2.5}$ concentrations at Chinese Research Academy of Environmental Sciences (CREAS) station, Guanyuan (GY) station and Dingling (DL) station; (b): Hourly variations of average PM$_{2.5}$ concentration over Beijing municipality.

④ Statistic metrics are listed in Table 1:

Table 1 Performance statistics of PM$_{2.5}$ concentration.

| Simulated Time Period | Stations | Obs. (µg·m$^{-3}$) | Sim. (µg·m$^{-3}$) | R | MB (µg·m$^{-3}$) | NMB (%) | NME (%) | MFB (%) | MFE (%) |
|---|---|---|---|---|---|---|---|---|---|
| 20:00 Nov. 17-22, 2012 | CREAS | 121.5 | 190.9 | 0.87 | 69.4 | 57.2 | 185.2 | 53.6 | 60.1 |
| | GY | 139.0 | 289.4 | 0.91 | 150.4 | 108.1 | 183.3 | 65.2 | 68.3 |
| | DL | 101.4 | 112.2 | 0.69 | 10.8 | 10.7 | 85.6 | 15.6 | 39.6 |

Notes: Mean bias: $MB = \frac{1}{n}\sum_{i=1}^{n}(Sim_i - Obs_i)$;

Normalized mean bias: $NMB = \frac{\sum_{i=1}^{N}(Sim_i - Obs_i)}{\sum_{i=1}^{N} Obs_i} \times 100\%$;

Normal mean error: $NME = \frac{1}{n}\sum_{i=1}^{n}\frac{|Sim_i - Obs_i|}{Obs_i} \times 100\%$

Mean fractional bias: $MFB = \frac{1}{N}\sum_{i=1}^{N}\frac{(Sim_i - Obs_i)}{(Obs_i + Sim_i/2)}$; Mean fractional error: $MFE = \frac{1}{N}\sum_{i=1}^{N}\frac{|Sim_i - Obs_i|}{(Obs_i + Sim_i/2)}$

• Section 4.1: There are several species and sectors that have emissions that contribute to $PM_{2.5}$ formation. Which emissions are considered in the presentation of the results here? In other works, how is $S_n$ defined? Are anthropogenic and natural sources included? What type of anthropogenic sources? Is it the total emission across all species? This is an essential missing detail. The results have little scientific or policy relevance in current form, given that they are only presented in terms of local vs nonlocal sources (a point for which use of an adjoint model would be overkill).

Response: Thank you for pointing this vague presentation out.

In the forward model processes, the emissions in GRAPES-CUACE include both anthropogenic and natural sources. Anthropogenic sources are constructed by Cao et al. (2011) based on statistical data from government agencies for the year 2007. They are the hourly gridded off-line emissions intensity for 32 species including black carbon (BC), organic carbon (OC), sulphate, nitrate, fugitive dust particles, in addition to 27 gases. Emission source types include biomass combustion, residences, power generation, industry, transportation, livestock and poultry breeding, fertilizer use, waste disposal, solvent use, and light industrial product manufacture (Cao et al., 2011). Natural sources are calculated with the parameterizations of natural sea salt and natural sand/dust emissions in the model.

In the backward GRAPES-CUACE aerosol adjoint model, as CAM (Canadian Aerosol Model) contains major aerosol processes (generation, hygroscopic growth, coagulation, nucleation, condensation, dry deposition/sedimentation, below-cloud scavenging, aerosol activation and chemical transformation of sulphur species in clear air and in clouds) in the atmosphere, the GRAPES–CUACE aerosol adjoint model is capable of coupling major aerosol processes in the atmosphere into its simulations of the sensitivities of the objective function to primary aerosol sources. Therefore, $S_n$ in the adjoint sensitivity coefficients $(\partial J/\partial S_n) \cdot S_n$ contains: black carbon (BC), organic carbon (OC), sulphate, nitrate, fugitive dust particles. Detailed description of emission sources and the definition of $S_n$ is added in section 3.2 & 3.3 (underlined in the manuscript). Spatial distribution and hourly variation of the primary $PM_{2.5}$ sources are also presented in section 3.3 (underlined in the manuscript).

To take advantage of the adjoint model, we subdivide local and nonlocal sources impacts into impacts from local Beijing and each surrounding province of Beijing: Beijing, Tianjin, Hebei, Shanxi and Shandong. A new section, section 4.4 is added to analyze the 'sensitivity coefficients of surface $PM_{2.5}$ concentration peaks in Beijing to primary emission sources from local Beijing and each of the surrounding provinces (Figure 9)'. Section 4.4 is in red.

[Figure]

Figure 9. Sensitivity coefficients of surface PM$_{2.5}$ concentration peaks in Beijing to primary emission sources from local Beijing and each of the surrounding provinces. The left and right panels correspond to PM$_{2.5}$ concentration peaks at 05:00 LT and at 23:00 LT on 21 November 2012 respectively. (a–b) illustrate hourly instantaneous sensitivity coefficients to emission sources from local Beijing, Hebei province, Tianjin city, Shanxi province and Shandong province. (c–d) show the time-integrated sensitivity coefficients to local and surrounding provincial emission sources. (e–f) are the contribution ratios of emission sources from each surrounding province to PM$_{2.5}$ concentration peaks.

• 8.9-11: This statement hasn't really been demonstrated. To use the adjoint sensitivities to "reproduce" the air pollution episode, one would need multiply the time series of sensitivities by the time series of emissions and show that their product matches the observations. This has not been done, nor would it likely work owing to nonlinearities.

Response: Thank you for pointing this out. The word "reproduce" is incorrectly used here. The authors want to convey that the GRAPES-CUACE aerosol adjoint model is capable of estimating the sensitivity of concentration to emission sources by **propagating a perturbation in concentration backward** while taking meteorological and chemical processes into consideration.

Therefore, this statement is changed to: This indicates that the GRAPES–CUACE aerosol adjoint model is capable of estimating the sensitivity of concentration to emission sources by propagating a perturbation in concentration **backward in time**, while incorporating meteorological and chemical processes. This presentation is at the end of section 4.2 and is in red.

Claims of efficiency are also implied but not quantified. A single adjoint model integration is often several times (2-10) slower than a normal forward model integration. Thus what is the overall computational savings of their approach here over forward methods, given the size of N and M, quantitatively?

Response: We checked the simulation record and randomly selected five simulation runs for both the

forward and adjoint models. As a result, a single adjoint model integration costs about 2 times computational time that of a normal forward model. For the forward GRAPES-CUACE modeling system, the computational time for integrating 288 steps (1 day) is about 220 min (215 min, 226 min, 226 min, 212 min and 222 min for each selected run). For the GRAPES-CUACE adjoint simulation, the computational time for integrating 288 steps backward is approximately 456 min (448 min, 464 min, 476 min, 443 min and 448 min for each selected run). Besides the high efficiency in computational time of the adjoint model illustrated above, the forward emission sources sensitivity analysis requires repeated adjustment of emissions intensity, which also requires tedious work.

However, adjoint method saves computational time at the expense of large computer storage for state data saving. Despite of it, the adjoint method is still a powerful tool in detecting sensitive emission sources of pollution episodes.

Example of the overall computational savings for the adjoint method over the forward sensitivity calculation: When calculating the changes of a scalar function (concentration) to N source emissions scenarios, N+1 times of forward integration is needed. If using the adjoint method, one forward integration (for state data saving) and one adjoint integration can achieve the sensitivities of the scalar function to the N emission sources at every integration step, much more sensitive information than the forward sensitivity analysis can achieve. Here the forward method costs (N+1)*t/2*2t=(N+1)/4 times computational time of that of the forward sensitivity estimation.

• Fig 6(e) and (f) are good to know, but they are somewhat of a waste of an adjoint model. If the only interest was in the separation between "surrounding" vs "local" emissions of all $PM_{2.5}$ precursor emissions, this could have been achieved with only 3 forward model integrations (adjoint not needed). So the authors haven't really brought out the strength of their results to provide insight into spatial attributions beyond these two regions. Pie chart showing the influence by province, species, and sector would be much more interesting, and would start to approach a level of detail unobtainable without use of an adjoint model.

Response: Thank you very much for your valuable suggestion. We then divided emission sources influence on the average hourly $PM_{2.5}$ peaks over Beijing municipality into local Beijing and provinces around Beijing (Hebei province, Tianjin city, Shanxi province and Shandong province). We analyzed the hourly instantaneous sensitivity coefficients of emission sources from each province, their corresponding time-integrated series and the overall contribution proportions of emission sources from each province to the $PM_{2.5}$ concentration peaks. Pie charts showing the influence by province are added in Figure 9e and 9f. This part is added in a new section (section 4.4) in the manuscript.

• Fig 8: Defining these ratios based on the area of the regions is not the best idea. It would be better to define the ratios based on the magnitude of the emissions in the different regions, since emissions intensity per unit area is not uniform.

Response: Following the referee's comments, we further defined an 'Emis-intense' region based on the magnitude of the emissions (Fig. 11), and compared its SC/PC (emission sources contribution ratios to peak $PM_{2.5}$ concentration peak), S/F(effect) (the ratio of sensitivity coefficients over Emis-intense to the sensitivity coefficients over HuaB, that is the sources contribution effects) and the S/F(efficiency) (the the ratios of sensitivity coefficients per unit area over Emis-intense to sensitivity coefficients per unit area over HuaB, that is the sources contribution efficiency). After comparison of the emission sources influence from Emis-intense, HuaB-sens and HuaB, we draw to the conclusion that controlling air pollutant sources from adjoint sensitive emission regions have better effects and higher efficiency than controlling emission sources from emissions intensive regions.

[Figure]

| Regions | Number of grid cells | Sensitive area ratios (%) |
|---|---|---|
| HuaB-sens | 18 | 10.2 |
| HuaB | 176 | |
| BJ-sens | 6 | 60.0 |
| BJ | 10 | |
| Emis-intense | 18 | 10.2 |

Figure 11. Domain definition of Huabei (HuaB, in red dot-dashed frame), Beijing (BJ, in black solid frame), sensitive Beijing (BJ-sens, red shaded), sensitive Huabei (HuaB-sens, red shaded and blue shaded) and emission intensive (Emis-intense, in pink solid frame) regions.

Statistical ratios are added in Table 2 and Table 3 in red. Corresponding analysis are added in section 4.5 in red.

In addition, there are reasons why regions are defined based on the adjoint results: to assess the adjoint sensitive source zone on decreasing $PM_{2.5}$ concentration over Beijing and to compare the adjoint results with the Models-3/CMAQ assessments. Therefore, through all comparisons in section 4.5, we reach the conclusion that narrowing emission sources reduction scope to sensitive source zones, rather than emission intensive regions, prior to unfavorable meteorological conditions can effectively decrease $PM_{2.5}$ concentration and improve the efficiency of $PM_{2.5}$ reduction measures.

• 10.25-28: Table 1 and the argument based on area isn't a great method, as discussed above. And I'm sorry but Table 2 and surrounding discussion just does not make much sense, and requires further clear explanation of what is being presented. What is the importance of the ratio SC/PC? This needs to be explained. What are the percent values percentages of? Do these sum to 100% in some manner? Lastly, comparison to results of Zhai et al. (2016) appears to be entirely qualitative, and no clear summary of how the two compare quantitatively is provided.

Response: SC/PC is the ratios of the time cumulative sensitivity coefficients to peak $PM_{2.5}$ concentration, and reflects the reduction ratios of peak $PM_{2.5}$ concentration due to absence in emissions over different regions and during different periods, that is the emission sources contribution ratios to peak $PM_{2.5}$ concentration. They do not sum to 100% as they are the primary particle sources of $PM_{2.5}$. Further clear explanations including the significance of SC/PC, S/F(effect) and S/F(efficiency) are added in section 4.5 in red. Comparisons with results of Zhai et al. (2016) are quantitatively compared by citing quantitative values from the Models-3/CMAQ assessments in the analysis of section 4.5 as well as in Table 2 and Table 3.

3. **Minor comments**

• 2.2: The first sentence is a bit vague, and should be clarified. Adjoint models are efficient for some types of sensitivity calculations, but not all. They are also efficient in terms of wall-time, but not necessarily in terms of memory or i/o.

Response: The adjoint model's efficiency is reflected by its short computational time costs in calculating the sensitivity of a scalar factor with respect to a large number of (input) parameters.

The first and second sentences are revised as:

The atmospheric chemistry adjoint model, developed on the basis of an atmospheric chemistry model according to the adjoint theory, can efficiently calculate sensitivities of a few variables or metrics with respect to a large number of (input) parameters (Sandu et al., 2005; Hakami et al., 2006).

• 2.14-19: This brief overview of "current" applications of adjoint modeling in atmospheric chemistry isn't a great fit for this paper, as it doesn't cover the first works in this area, historically, nor is it limited to only the latest works. Also, in attempting to cover all applications of adjoint model, the authors touch upon several areas ($O_3$, $CO$, etc.) that aren't directly related to the topic of $PM_{2.5}$. I suggest the authors instead consider a more detailed overview of previous works, but one that is more narrowly limited in terms of scope, possibly to just sensitivity studies of $PM_{2.5}$.

Response: Thank you for your suggestion. We revised this paragraph by previous research on the sensitivity studies of $PM_{2.5}$. Detailed responses including changes in the manuscript are presented in the second major comment.

• 3.2: There is a second paper by the same group using adjoint modeling to investigate sources of $PM_{2.5}$ in Beijing during the APEC period.

Response: There is a paper using the adjoint of GEOS-Chem to investigate sources of $PM_{2.5}$ in Beijing during the APEC period:

Zhang, L., J. Shao, X. Lu, Y. Zhao, Y. Hu, D. K. Henze, H. Liao, S. Gong and Q. Zhang (2016). "Sources and Processes Affecting Fine Particulate Matter Pollution over North China: An Adjoint Analysis of the Beijing APEC Period." Environmental Science & Technology 50(16): 8731-8740.

• 3.9: Could the authors clarify what is meant by "guidance on the enaction of dynamic environmental control policy"? What type of policy are they referring to (municipal? national? international?), and what is dynamic about such policy?

Response: 'Dynamic' is changed to 'flexible'.

• 4.13: Technically a first-order finite difference calculation would require N+1 forward model integrations.

Response: Thank you for pointing out this inexact presentation. We changed N to N+1 and added '(one base simulation included)' at the end of the sentence.

• 4.18: The theoretical equivalence of these approaches predates the work of Liu by many decades; I suggest the authors find a more fundamental reference. Also, it is typical to only cite PhD thesis (as opposed to peer reviewed literature) when absolutely necessary, which is not the case here.

Response: We replaced this reference by:

Marchuk, G., Mathematical Models in Environmental Problems, Elsevier Science Publication Co., Washington, 1986.

• 4.26: Adjoint sensitivities would only provide "exact" contributions for linear systems. However, $PM_{2.5}$ is formed nonlinearly, which needs to be addressed, or the interpretation and use of the adjoint sensitivities needs to be reconsidered.

Response: Thanks for pointing out. This presentation is changed to:

Therefore, we define the sensitivity coefficients in this study as: $(\partial J/\partial S_n) \cdot S_n$, which shares the same unit with the objective function, and can reflect the absolute changes in the objective function due to

perturbations in emission sources, thus making comparisons among emissions more convenient.

• 5.14: "Unequilibrated" is not the correct word here. Nonlinear?

Response: Here we want to refer to state variables whose values change with the integration of the forward model. When integrating the adjoint model backward, these saved values are input at each check point to ensure that the forward and the backward models are in the same chemical state. The word 'unequilibrated' is used by Henze et al. (2007). Sandu et al. (2005) describe it as 'the state $c(x, t)$ saved for all $t$'. To make it clear, we change this part in the manuscript as below, with changes in red:

Figure S3 shows the operational processes in this study. In order to ensure that the forward and the backward models are in the same chemical state, the forward GRAPES-CUACE model should be first integrated to save the unequilibrated variables (or called the state concentrations) in checkpoint files at the beginning of each external time step (Sandu et al., 2005; Henze et al., 2007). These saved variables were then input at each check point during the backward adjoint integration. To handle intermediate variables, this study adopted recalculation and stack storage (PUSH & POP) schemes.

• Section 3.2: In addition to the physical processes treated in this aerosol model, please also briefly review what chemistry is included, both in the aerosol and gas-phase, and how the thermodynamic partitioning of species across phases is modeled.

Response: The gas chemistry is based on the second generation of Regional Acid Deposition Model (RADM Ⅱ) mechanism with 63 gaseous species through 21 photo-chemical reactions and 121 gas phase reactions applicable under a wide variety of environmental conditions especially for smog (Stockwell et al., 1990). Aerosol processes is coherently integrated with the gaseous chemistry in CUACE (Zhou et al., 2012). Since the nitrates and ammonium formed through the gaseous oxidation are unstable and prone to further decomposition back to their precursors, CUACE adopts ISSOROPIA to calculate the thermodynamic equilibrium between them and their gas precursors (Zhou et al., 2012). Relevant model descriptions are added in section 3.2 (underlined in the manuscript).

• Section 3.3: Previous studies have shown that there are influences of emissions on PM$_{2.5}$ in your receptor cite from beyond the model domain considered here. Thus please explain how the influence from boundary conditions is tracked in the adjoint modeling.

Response: Both the forward and the adjoint model domain covered northeastern China, as shown in Fig. 4 (left). This domain is big enough to take pollutant regional transport into consideration. To better address this problem, we enlarged the cover area in the sensitivity coefficients distribution diagrams in Fig. 7 and Fig. 10. Moreover, we modified the color bars in Fig. 7 and Fig. 10, with values less than 1 amplified in the range of the color bars. Therefore, it is quite clear to see that the value of sensitivity coefficients decreased 2-3 orders of magnitudes to the edge of Fig. 7 and Fig. 10.

Meanwhile, we also enlarged the area when extracting emission sources sensitivity coefficients for the surrounding regions of BJ in section 4.3. Therefore, overall sources contribution ratios from surrounding provinces (add overall contribution ratios from each surrounding province in the pie chart in Fig. 9) are slightly increased compared with the last version of the manuscript.

We hope these changes in the sensitivity coefficients distribution diagrams can help to answer this comments.

[Figure]

**Figure 7. Time-integrated sensitivity coefficients of surface Beijing PM$_{2.5}$ concentration peaks to primary emission sources. (a–d): 1-h, 12-h, 24-h and 72-h integrated sensitivity coefficients for the 5:00 LT on 21 Nov. PM$_{2.5}$ concentration peak; (e–h): 1-h, 12-h, 24-h and 72-h integrated sensitivity coefficients for the 23:00 LT on 21 Nov. PM$_{2.5}$ concentration peak.**

[Figure]

**Figure 10. 24-h (a), 48-h (b) and 72-h (c) integrated sensitivity coefficients of surface PM$_{2.5}$ concentrations to primary emission sources in Beijing on 21 Nov. 2012.**

• 7.12: This is a more correct interpretation of adjoint sensitivity results which should be considered in the earlier descriptions.

Response: Thank you. Earlier relevant descriptions are revised to **emphasize** that the adjoint sensitivity reveals the changes in concentration due to small perturbations in emissions sources, in other words, the adjoint model estimates the incremental influence of specific sources on air quality attainment. Corresponding revisions are green shaded.

• Fig 5: It appears the emissions continue to spread by 72 hours of back integration. How then did the authors decide to stop the adjoint integration at 72 hrs? In other words, why did they not integrate

backwards further in time? The lifetime of aerosols can be much longer than 3 days, so integration of back to a week to 10 days may be necessary to capture all non-local influences.

Response: From the hourly instantaneous sensitivity coefficients variation in Fig. 8a and 8b and Fig. 9a and 9b, sensitivity coefficients decrease to negligible values (close to zero) after 72 hrs of adjoint integration. Meanwhile, from the time-integrated sensitivity coefficients in Fig. 8c and 8d and Fig. 9c and 9d, time-integrated sensitivity coefficients prone to constant values after 72-hr accumulation.

• Section 4.1: Please clearly define what is meant by "local" in this context. Is it just the single grid cell that contains the Beijing receptor cite?

Response: "Local" refers to the objective region that covers the whole Beijing municipality (both rural and urban Beijing). Definition is added as: local Beijing (the target region that covers the entire Beijing municipality).

• 9.24-26: The non-local contributions do get small after 72 hrs, but as shown in Fig 6(d), the cumulative sensitivities have yet to asymptote to a constant value, which would indicate that sensitivities from early than 72 hrs may still play some role, although small. Also, sensitivities may have transferred to the boundary conditions, as mentioned previously.

Response: As analyzed in section 4.3 and section 4.4, the 23:00 $PM_{2.5}$ peak was accumulated on the basis of the 05:00 $PM_{2.5}$ peak, and the maximal points on Fig 8b are actually due to the same sources for the maximal points on Fig. 8a. Therefore, the cumulative sensitivities in Fig. 6d (now changed to Fig. 8d) is expected to stop increasing soon afterwards.

**4. Corrections**

I started making grammatical corrections to the abstract, but stopped after only a few lines, as the entire manuscript needs substantial editing.

• 1.17: in detecting ! to detect

• 1.20: south to ! south of

• 1.21: at the south to ! to the south of

Response: Thank you very much for your revisions.

The above three errors are corrected. Besides, the whole manuscript is edited by the Enago (http://www.enago.com) English editing services and verified by the authors. The authors will also order the English copy-editing services from Copernicus.

**Other Changes in the manuscript:**

The title was changed to 'Detection of critical $PM_{2.5}$ emission sources and their contributions to a heavy haze episode in Beijing, China, using an adjoint model'.

Section 3.4 'Simulated haze episode' and section 3.5 'Objective function' are moved to section 4.1 'Simulated haze episode and objective function'. Former section 4.1, section 4.2 and section 4.3 are placed to section 4.2, section 4.3 and section 4.5, respectively.

Added analysis of Section 4.4 'Emission sources contributions from different provinces around Beijing to peak $PM_{2.5}$ concentrations' and comparisons of emission sources contribution from emission intensive regions and adjoint sensitivity regions are added in section 4.5, the abstract and the conclusion.

**Added References:**

Cao G L, Zhang X Y, Gong S L, et al. Emission inventories of primary particles and pollutant gases for China. Chinese Sci Bull, 2011, 56, doi:10.1007/s11434-011-4373-7.

Hakami, A., D. K. Henze, J. H. Seinfeld, K. Singh, A. Sandu, S. Kim, D. Byun and Q. Li (2007). "The adjoint of CMAQ." Environmental science & technology 41(22): 7807-7817.

Sandu, A., D. N. Daescu, G. R. Carmichael and T. Chai (2005). "Adjoint sensitivity analysis of regional air quality models." Journal of Computational Physics 204(1): 222-252.

Stockwell, W. R., Middleton, P., Change, J. S. and Tang, X. 1990. The second generation regional acid deposition model chemical mechanism for regional air quality modeling. J. Geophys. Res. 95, 16343_16376.

Zhai, S., X. An, Z. Liu, Z. Sun and Q. Hou (2016). "Model assessment of atmospheric pollution control schemes for critical emission regions." Atmospheric Environment 124, Part B: 367-377.

Zhou, C., S. Gong, X. Zhang, H. Liu, M. Xue, G. Cao, X. An, H. Che, Y. Zhang and T. Niu (2012). "Towards the improvements of simulating the chemical and optical properties of Chinese aerosols using an online coupled model − CUACE/Aero." Tellus B 64(0).

**To Short Comment #1**

Dear G. Tang,

Thank you very much for your comments. Responses to the comments are listed below in blue:

As for the regional and local contributions for the Beijing haze pollution, we also launched some observation studies, but were not verified by model simulations [G. Tang et al., 2015; Zhu et al., 2016]. After I read this paper, I was so glad that the simulated results were consistent with our findings. However, there were some tiny questions that I want to discuss with the authors:

a.   The authors investigated the haze pollution in Beijing based on the two peaks of $PM_{2.5}$ concentrations, and the $PM_{2.5}$ concentrations were peaked on 05:00 BT and 23:00 BT Nov. $21^{st}$ 2012 respectively. However, these two moments are probably the high pollutant concentration periods originally due to some local emissions (such as the diesel cars). Does it necessary to divide the period so detail? I suppose if it was better to divide the haze episode into two stages, the first stage (from Nov. 19 to 20 2012) may be defined as the early polluted period and the second stage (included the two peaks) may be defined as the heavy polluted period. And then to quantify the regional transport and local contributions.

Response: The high $PM_{2.5}$ pollution on Nov. $21^{st}$ 2012 over Beijing is the result of both local and surrounding pollution sources, illustrated in section 4.5 when average $PM_{2.5}$ concentration on Nov. $21^{st}$ 2012 over Beijing is set as the objective function and that the relatively more sensitive regions spread over Beijing and surrounding Hebei, Shanxi and Shandong provinces (Fig. 10). **Under the high pollution background on Nov. $21^{st}$ 2012, two hourly $PM_{2.5}$ peaks occur.** These two peaks are resulted mainly from the diurnal variation of (planetary boundary layer) PBL with the development of vertical mixing after sunrise for diluting pollutants (added in section 4.1). Meanwhile, from Fig. 7b and 7f, we can see that the 12-hour cumulated sensitivity coefficients mainly concentrated over Beijing, especially for Fig. 7f, reflecting that local source impacting on the pollution peaks dominate 12 hours ahead of the peaks. However, analyzing Figs. 7c, 7d, 7g and 7h, we can see that

surrounding sources emitted 2-3 days ahead have obvious impacts on these two peaks. Besides, analyzing the time-cumulated sensitivity coefficients series (Fig. 8c and 8d) along reversed time sequence, we can see that for both PM$_{2.5}$ concentration peaks, the dominant emission source areas transform from the local to the surroundings. **To sum up, these two peaks were due to local impacts at the same day (Nov. 21$^{st}$), but are influenced by local and surrounding sources during the whole pollutants accumulation and polluted episode (18$^{th}$-21$^{st}$ Nov. 2012).**

Thank you for your suggestion. We expect it will be an interesting and worthy job to divide the haze episode into two stages as you describe, and we will try to conduct such a simulation in our later work. Here we want to illustrate the applicability of the GRAPES-CUACE aerosol adjoint model in tracking influential PM$_{2.5}$ emission source regions and time periods in detail. Especially, to divide the period in detail, the adjoint results clearly illustrate the difference of these two peaks influential sources spatially and temporally, as the first peak is the result of local and surrounding sources accumulation 2-3 days ahead of it, and the second peak is accumulated on the basis of the first peak with dominant influence from local emissions on Nov. 21$^{st}$.

b.  Does the obvious periodic fluctuation of hourly sensitivity coefficient of surrounding emissions has some relationship with the mountain-valley winds [Guiqian Tang et al., 2016]?

Response: Yes, the mountain-valley winds have impacts on the periodic fluctuation of hourly sensitivity coefficient of surrounding emissions. When there is no strong weather system, obvious mountain and valley winds appear in the plain area of Beijing-Tianjin-Hebei, and a convergence line shows at the surface layer. During the daytime, valley winds strengthened while mountain winds weakened. During the nighttime, however, mountain winds strengthened while valley winds weakened. Therefore, the convergence line vibrates periodically in the north-south direction, and pollutants accumulated nearby and to the south of the convergence line.

Tang, G., et al. (2016), Mixing layer height and its implications for air pollution over Beijing, China, Atmospheric Chemistry and Physics, 16(4), 2459-2475, doi:10.5194/acp-16-2459-2016.

Tang, G., X. Zhu, B. Hu, J. Xin, L. Wang, C. Münkel, G. Mao, and Y. Wang (2015), Impact of emission controls on air quality in Beijing during APEC 2014: lidar ceilometer observations, Atmospheric Chemistry and Physics, 15(21), 12667-12680, doi:10.5194/acp-15-12667-2015.

Zhu, X., G. Tang, B. Hu, L. Wang, J. Xin, J. Zhang, Z. Liu, C. Münkel, and Y. Wang (2016), Regional pollution and its formation mechanism over North China Plain: A case study with ceilometer observations and model simulations, Journal of Geophysical Research: Atmospheres, 2016JD025730, doi:10.1002/2016JD025730.

**Added Reference:**

[revised manuscript text omitted]

---

## Author Response (AR2)

Responses to referee #3 and referee #2 on "Detection of critical $PM_{2.5}$ emission sources and their contributions to a heavy haze episode in Beijing, China, using an adjoint model", and marked-up version of the manuscript.

Dear referees,

Thank you very much for your valuable comments. Revisions in the marked-up manuscript version are in red. This document is organized as follows: the referees' comments are in black and responses to the comments are in blue.

**To Referee #3**

**Suggestions for revision or reasons for rejection (will be published if the paper is accepted for final publication)**

The revised manuscript by Zhai et al. has addressed many of the previous reviewer concerns, particularly with regards to expanding their work to better demonstrate the value of their adjoint modeling. They also include additional comparison to observations, and expanded comparison to previous work. Still, I found the description of the treatment of secondary aerosol species such as sulfate and nitrate very confusing (not clear if these are even modeled as secondary species), the model evaluation showed significant model overestimation, and the presentation of adjoint sensitivities for the purpose of source attribution is not considered or presented carefully. I believe addressing these issues constitutes another round of major revisions, after which this article will be more suitable for publication in ACP.

Major comments:

A large fraction of the $PM_{2.5}$ concentrations in the pollution episodes studied here consist of secondary inorganic aerosol. However, there are several issues related to the treatment of such species that need to be clarified:

1. It is not clear if ISORROPIA and it's adjoint were used in this study. The authors state that the GRAPES-CUASE aerosol adjoint includes CAM, citing Gong et al., 2003, but that version of CAM did not use ISORROPIA, which wasn't added until Zhou et al. (2012). This needs to be clarified. Developing the adjoint of ISORROPIA is an extremely challenging task, the subject of an entire manuscript by developers of ISORROPIA (Capps et al., 2012). If the authors did include ISORROPIA, did they develop their own version, or use the one from Capps 2012 (ANISORROPIA)?

Response: In this study, the forward GRAPES-CUACE modeling system is an online coupled atmospheric chemistry modeling system, and CUACE is an atmospheric chemistry modeling system independent of meteorological and climate models. CUACE adopts CAM (Canadian Aerosol Module) as its aerosol module, the RADM II mechanism as its gaseous chemistry and ISORROPIA to calculate the thermodynamic equilibrium between nitrates (and ammonium) and their gas precursors.

The current version of the GRAPES-CUACE aerosol adjoint model includes ①the adjoint of CAM, ② the adjoint of three interface programs that pass meteorological variable values to the chemistry processes, and ③the adjoint of the aerosol transport processes. Since the adjoint of gaseous chemistry and the adjoint of ISORROPIA are now under development, the current version of the aerosol adjoint model doesn't include the adjoint of ISORROPIA yet. The current version of the GRAPES–CUACE aerosol adjoint model is capable of coupling major aerosol processes in the atmosphere into its simulations of the sensitivities of the objective function to **primary** aerosol sources. **These are clarified in the first and second paragraphs of Section 3.2 (P6 Line 147-168).**

2. This issue is further confused by the fact that (Section 3.3) the authors state the model includes emissions of sulfate and nitrate. But sulfate and nitrate are species that are formed secondarily in the atmosphere, and the emitted species should be $SO_2$ and $NO_x$ (and $NH_3$). If instead the model somehow approximates inorganic $PM_{2.5}$ as being a primary, rather than secondary, aerosol, the validity and accuracy of this assumption needs to be discussed, as it would be in conflict with most regional air quality models. It might though explain the very high bias of the GRAPES-CUASE model compared to the observations.

Response: Thank you for pointing this out. In GRAPES-CUACE, sulfate and nitrate come from both primary emissions and secondary formation from $SO_2$ and $NO_x$. The sulfate and nitrate emissions here refer to **primary** sulfate and nitrate particle sources. **To make the emissions description clear, we added 'primary' before OC, sulfate and nitrate in Section 3.2 (P6, line 167), Section 3.3 (P6, line 186) and Section 4.5.**

We agree with the referee's comment on inorganic $PM_{2.5}$. In the revised manuscript, we clarified that the emission inventory also included $SO_2$ and $NO_x$ (and $NH_3$) besides sulfate and nitrate in the forward GRAPES-CUACE modeling. Secondary aerosol formations is an important process of atmospheric physics and chemistry with large uncertainties, based on the current understanding on atmospheric environment. Generally, three factors controlling the discrepancies in air quality modeling are 1) air pollutant emissions, 2) physical and chemical processes in the atmosphere and 3) meteorology especially in the boundary layer (An et al., 2013; Cheng et al., 2016; Wang et al., 2015a; Wang et al., 2016). Overestimation of $PM_{2.5}$ in this study might be attributed to the uncertainties of these three factors in model. **Details are explained in the last paragraph in section 3.3.**

3. Even if the authors did include secondary inorganic aerosols in their adjoint modeling, the accuracy of their adjoint model for estimating the contribution of emissions perturbations to $PM_{2.5}$ concentrations is undermined by two issues: adjoint code accuracy and nonlinearity of $PM_{2.5}$ formation (even within a perfect adjoint code).

3a Adjoint accuracy: While trying to investigate this in the present paper and previous manuscripts, I came to realize that the GRAPES-CAUSE model has not demonstrated the accuracy of the emissions sensitivity coefficients, which is particularly important if the authors are calculating the sensitivity of secondary aerosol species (e.g. nitrate) with respect to precursor emissions ($NO_x$). Maybe they don't include secondary aerosols in their work (not clear, see previous question), but if they do they need to provide detailed evaluation of adjoint sensitivities to demonstrate it is working. In An et al. (2016) they present results for the passing the Lagrange condition for a particular model configuration, but the details of that configuration were not clear (how long was the simulation? what species were included in xrow?), and results specifically relating to aerosol thermodynamics were not reported. They only showed tests with respect to concentrations, not emissions. Given the difficulties identified in Capps et al. (2012) with evaluating the extremely nonlinear, discontinuous ISORROPIA behavior using finite difference approaches, I'm skeptical as to the accuracy of the aerosol adjoint sensitivities for $SO_2$, $NO_x$ and $NH_3$. It would helpful if they could show some finite difference tests relating estimates of the change in $PM_{2.5}$ owing to perturbations of emissions of $SO_2$, $NO_x$ and $NH_3$ in particular grid cells, for perturbations of various magnitudes. Not only would these tests evaluate the accuracy of the model, they would also help determine the range of perturbations over which the adjoint sensitivities can be interpreted as source contributions, as discussed below.

Response: The forward GRAPES-CUACE modeling system considers secondary $PM_{2.5}$ formations from precursor gases. However, in the GRAPES-CUACE aerosol adjoint model, the adjoint of gaseous chemistry is now under development. Therefore, gaseous precursor emissions were not considered

while calculating the sensitivity of aerosols with respect to emissions. **Clarifications have been made in Section 3.2.**

In An et al. (2016), the tangent linear (TLM) and the adjoint model were tested according to **rigorous mathematical derivations**, which is **a different method** from the finite difference test that were implemented in the validation of the adjoint of CMAQ (Hakami et al., 2007), the adjoint of GEOS-Chem (Henze et al., 2005), etc. **Our clarifications have been made in Section 3.2, Paragraph 2.**

In our validation (An et al., 2016), the tangent linear and the adjoint model integrate at a time step of 30 min. The tangent linear model was integrated for 10 steps, and the adjoint model was integrated for 12 steps (6 hours). In addition, in the work of An et al. (2016), every input variable in the model has passed the validation, but we only illustrate two variables (*xrow* and *rhop*) as examples due to limited space of the paper. CAM involves six types of particles – sulfate, OC, BC, nitrate, sea salt, and soil dust – which are divided into 12 sections using the multiphase multicomponent aerosol particle size separation algorithm and were represented by *Xrow* in the model. *Rhop* represents particle wet radius. As mentioned before, although we did include ISORROPIA in the forward GRAPES-CUACE model while simulating $PM_{2.5}$, we didn't include ISORROPIA adjoint in the current version of the GRAPES-CUACE aerosol adjoint model. Therefore, aerosol adjoint sensitivities for $SO_2$, NOx and $NH_3$ are not considered in this study.

Following your suggestion, we further verified the adjoint code by comparing the adjoint sensitivities with the finite difference results:

[Figure]

Figure 1 Comparisons of the adjoint sensitivity coefficients (red line with triangle symbols) and the finite difference results (blue line with circular symbols) for $PM_{2.5}$ primary emission reduction ratios at 5%, 10%, 20%, 30%, 50%, 70% and 90% over simulation domain for the Nov. 21 05:00, 2012 $PM_{2.5}$ peak.

Figure 1 shows the comparisons of the adjoint sensitivity results and the finite difference results. From Fig.1, we can see that $PM_{2.5}$ concentration and its primary emission sources have a linear relationship. Because of the linear relationship between $PM_{2.5}$ concentration and its primary emission sources, the magnitude of perturbations will not influence the representative of the adjoint sensitivities when comparing contribution proportions of emission sources from different regions. However, if using the adjoint sensitivities to represent the absolute emission source contributions, errors will increase with the increase of perturbations. In Fig. 1, we can see that the adjoint sensitivity results are close to the finite difference results, so that the adjoint sensitivity coefficients are likely to be valid over $PM_{2.5}$ primary

emission reduction ratios from 5% to 90%, or at least over a modest range of emissions perturbations commensurate with typical emissions abatement strategies (10-30%). **A new paragraph in red characters is added in Section 3.1 to clarify this, and Fig. 1 is added as Fig. S3 in the supplement.**

3b Nonlinearity: For nonlinear models of PM$_{2.5}$ formation, adjoint sensitivities can not be directly equated to source contributions, even if they are correctly coded. In my original response, I requested the authors clarify this issue in their description of the use of adjoint sensitivities, but their modifications in this regard fell short. The abstract presents adjoint sensitivities as contributions, with no consideration for this issue. The first paragraph of the introduction and several other locations highlights the benefits of adjoint sensitivity analysis without recognizing the limitation of this approach for evaluating complete source contributions. The authors need to think more carefully about how their results can be considered. Over what range of emissions perturbations, to which species, are their sensitivities accurately representative? Did they test this? Or there are other studies in the literature to reference here? Does this impact their comparison to the forward modeling study of Zhai et al (2016)?

**Response**: Thank you very much for your comments and we have revised the manuscript following your suggestions. Adjoint sensitivities are the tangent linear gradients of the objective function to emissions (or initial concentrations), and are equivalent to the increments of the objective function owning to small perturbations (Henze et al., 2007 and 2009). For nonlinear models of PM$_{2.5}$ formation, the state of the system might change under large perturbations in input variables. Therefore, we are very caution to use sensitivities to represent emission sources contributions for large changes in emissions. **The limitation of the adjoint approach for evaluating complete source contributions is added in the first paragraph of the introduction. A new paragraph marked in red color is added in Section 3.1 to clarify this, and Fig. 1 is added as Fig. S3 in the Supplement.**

In this study, we only consider adjoint sensitivity of the peak PM$_{2.5}$ concentration to its primary emission sources, and the peak PM$_{2.5}$ concentration have a linear relationship with its primary particle sources in GRAPES-CUACE through finite difference tests (Fig. 1). As the adjoint results are close to the finite difference results, adjoint sensitivities are likely to be representative over PM$_{2.5}$ primary emission reduction ratios from 5% to 90%. As discrepancies between the adjoint sensitivity results and the finite difference results increase with the increase of emission reduction ratios, the adjoint sensitivity coefficients at least are representative over a modest range of emissions perturbations commensurate with typical emissions abatement strategies (10-30%). **Clarifications are added in Section 3.1 in red.**

There are several literatures that considered the representative of the adjoint sensitivities in source attribution. **It turns out that the adjoint model is overall a promising tool for examining the dependence of aerosol concentrations on emissions** (Henze et al., 2007).

Take three typical researches for example:

① In the work by Henze et al. (2007), the authors explore the robustness of the aerosol sensitivities with respect to the magnitude of the emissions. It turns out that while individual sensitivities may be valid only over limited range, the sensitivity field as a whole appears fairly robust. ② Similarly, in another work by Henze et al. (2009), they point out that while adjoint sensitivity analysis is not strictly a method for source apportionment, **it does have several attractive aspects for estimating the incremental influence of specific sources on air quality attainment**. Then the authors present that the adjoint sensitivities are likely to be valid over a modest range of emissions perturbations commensurate with typical emissions abatement strategies (10-30%). ③ Zhang et al. (2015) implement the adjoint method in source apportionment of particulate matter pollution over North China. They use the

magnitudes of adjoint sensitivities to **approximately** represent source contributions to $PM_{2.5}$ concentrations. They also examined that the sum of the adjoint sensitivity accounts 86% of the simulated mean $PM_{2.5}$ concentration, and point out that the discrepancy is mainly attributed to the nonlinear response of $PM_{2.5}$ to emissions.

In this study, several factors might impact the comparison of the adjoint results and the forward modeling study of Zhai et al (2016), for example, the differences in emissions accuracies, and the differences in meteorological and chemical mechanisms in MM5-CMAQ and GRAPES-CUACE models. Except for the above mentioned factors, compared with the forward modeling analysis which considers both primary and precursor sources of $PM_{2.5}$, the deficiency of the adjoint analysis in this study is that we didn't include $PM_{2.5}$ precursor emissions. Nevertheless, through comparison, we find that **the two modeling approaches are highly comparable in their assessments of atmospheric pollution control for critical emission regions. Overall, the adjoint sensitivities of peak $PM_{2.5}$ concentration to primary particulate emissions using the GRAPES-CUACE aerosol adjoint model can provide valuable reference on evaluating emission impacts on pollutant concentrations and air quality control. Clarifications are added in Section 4.5 in red.**

4. I appreciate the additional content on model evaluation. Still, the performance here seems to be presented in a somewhat overly optimistic light. The authors did not deeply consider reasons why their model seems to overestimate $PM_{2.5}$ peaks by >200 $\mu g/m^3$. They mention the possibility of model resolution error, but the biases seem large and typically in the opposite direction of model resolution bias, although if the authors know that the measurement sites are located in locations that are more pristine than the surrounding areas, this should be stated. It seems that other sources of error, such as uncertainties in emissions, or treatment of all $PM_{2.5}$ as primary species (see other comments with regards to sulfate, nitrate, etc.) may be to blame. Lastly, and even more importantly, the authors should figure out how these types of model performance deficiencies impact their interpretation of adjoint sensitivities. Can these be used to estimate uncertainties in their sensitivity analysis as well? The authors also did not address my previous specific questions about the ability to simulate such peaks without inclusion of heterogenous chemical reactions recently identified (Wang et al., PNAS, 2016, doi:10.1073/pnas.1616540113; Cheng et al., Science Advances, 2016, doi:10.1126/sciadv.1601530) — I would have thought their model would be low biased, not high biased. Please explain.

Response: Thank you for your comments. In the revised manuscript, we clarified that the emission inventory also included $SO_2$ and NOx (and $NH_3$) besides sulfate and nitrate in the forward GRAPES-CUACE modeling. Secondary aerosol formations are important processes of atmospheric physics and chemistry with large uncertainties, based on the current understanding on atmospheric environment. Generally, three factors controlling the discrepancies in air quality modeling are 1) air pollutant emissions, 2) physical and chemical processes in the atmosphere and 3) meteorology especially in the boundary layer (An et al., 2013; Cheng et al., 2016; Wang et al., 2015a; Wang et al., 2016). Overestimation of $PM_{2.5}$ in this study might be attributed to the uncertainties of these three factors in model. As the following analysis mainly focus on the variations and contribution proportions of emission sources over different regions, adjoint sensitivity analysis are not significantly affected by overestimation of $PM_{2.5}$ and these modeling results can be considered reliable. **Clarifications are made in Section 3.3**.

The recently identified heterogeneous chemical reactions (Wang et al., 2016; Cheng et al., 2016) are missing sources of sulfate formation during high $PM_{2.5}$ episodes in traditional chemical models (e.g.: the current GRAPES-CUACE modeling system). In future model development, we should try to include

recently identified chemical reactions in chemical models to improve the simulation performance. We suppose other deficiencies in model simulation and data observation mentioned above might offset the underestimation here.

5: Most of the sensitivity results appear to be integrated across all sectors and species, and reflect the influence of all $PM_{2.5}$ emissions. It would be useful, however, to know the extent to which different species and sector contribute as well, as policies generally target particular species and sectors. It would also show another benefit of the adjoint modeling approach, which provides information across species and sectors without any additional model runs.

Response: Thank you very much for your suggestion. Since the quantifying of relative contributions of local emission and regional transport is a critical issue for air pollution management and is still in debate, in this study, we focus on detection of critical emission source regions with the help of adjoint sensitivity analysis. To know the contribution of different species and sectors would be useful to policy making, and we have worked on emissions contributions from different species. Due to limited space of the paper, this part of study is not included in this work.

Minor comments:

- Abstract: missing a period.

Response: The Global–Regional Assimilation and Prediction System coupled with CMA Unified Atmospheric Chemistry Environment (GRAPES–CUACE) aerosol adjoint model was applied to detect the sensitive emission sources of a haze episode in Beijing **during 19–21 November 2012**.

- ISORROPIA is mis-spelled.

Response: Revised.

- I was curious about their use of the word "unequilibrated", so I checked Henze et al. (2007). In the latter, this word was used specifically with regards to the input variables for the aerosol thermodynamic equilibrium calculation. It does not make sense to use in the current manuscript as written — I suggest "…integrated to save the model state variables (concentrations) in checkpoint files…".

Response: Thank you. We have revised "unequilibrated data" to "model state variables" according to your suggestion.

Abstract, last sentence: This is a key point, but it needs rewording. Suggest: "…controlling air pollutant sources from regions identified using adjoint sensitive analysis would lead to greater $PM_{2.5}$ reductions per source control than from regions with the greatest emission intensities."

Response: Thank you. We have reworded this sentence according to your valuable suggestion.

Introduction: The phrase "adjoint sensitive zone" needs to reworded, or at least defined, as "zones with maximum adjoint sensitivities" or similar.

Response: Thank you. "Zones with maximum adjoint sensitivities" is surely a clearer description of zones that are detected by adjoint sensitivity coefficients. We have reworded "adjoint sensitive zone" to "zones with maximum adjoint sensitivities".

Fig 5: Why is there also not a black line connecting the triangles showing the observation points? This would make it easier to compare the patterns of the observations as compared to the model.

Response: Black lines connecting triangles are added in Fig. 5.

Section 4.3: Although now defined earlier, as requested, it might be good to restate here what is defined as "local".

Response: Restated.

**Response**: Clearer content of Figure 1 are provided in the manuscript. Enlarged content of Figure 1 are then deleted in the Supplement.

Yes, both red and blue shaded region denotes sensitive Huabei. To make it clear, we revised "red shaded and blue shaded" to "both red and blue shaded" in the caption of Figure 11.

[revised manuscript text omitted]

---

## Author Response (AR3)

Responses to 3 referees on "Detection of critical PM$_{2.5}$ emission sources and their contributions to a heavy haze episode in Beijing, China, using an adjoint model", and marked-up version of the manuscript.

Dear referees,

Thank you very much for your valuable comments and suggestions. Revisions in the marked-up manuscript version are in red.

**Reply to Referee #1**

Response: Thank you very much for your recognition.

**Reply to Referee #2**

**General comments**

*This manuscript practices the application of an adjoint model of GRAPES-CUACE developed by An et al. (2016) in tracking PM$_{2.5}$ influential emissions during a typical haze episode in Beijing, China's massive capital. As a kind alternative source apportionment technique, the mode using the gradient of the objective function to emissions, is able to quantitatively present the emission contribution to PM$_{2.5}$ concentrations. The results are impressive, and the study is helpful to understand how the regional emissions impact local PM$_{2.5}$ levels, especially in Beijing. The subject provides an alternative way to conduct sensitivity analysis of emissions to concentrations. It is meaningful to air quality agencies to develop emission control measures and emission reduction plan. However, the basic assumption of "the primary particulate emission sources and PM$_{2.5}$ concentrations have an approximately linear relationship" works only under the condition of calm weather. This assumption, from my point of view, may be very questionable if the enhanced inflow and high background particulate levels due partly to strong transport.*

*The quality of the manuscript including all figures has been significantly improved as the result of revised version. I recommend it be accepted to publish in Atmospheric Chemistry and Physics via minor revisions.*

Response: Thank you very much for your valuable comments.

We agree with the comments that the primary particulate emission sources and PM$_{2.5}$ concentrations have an approximately linear relationship" works only under the condition of calm weather. The relationship between PM$_{2.5}$ and their primary emission sources are complicated by different weather conditions. Further study is required to investigate the relationship with adjoint sensitivities' representation of emission source contribution under different weather conditions.

In the revised manuscript, we have added the above-mentioned discussion in Section 5 Conclusions (lines 467-469)

**Major comments**

*The case study is based on single mechanism (CAM+RADM2) for aerosol and gas phase. I am wondering if different mechanisms affect PM$_{2.5}$ concentrations. The conclusion may become stronger if the authors could provide the level of uncertainties of the results caused by different mechanisms.*

Response: We agree with the referee that the conclusions will become stronger if providing the level of uncertainties of the results caused by different mechanisms. We are now coupling the CB-IV mechanism in the GRAPES-CUACE forward model, and embedding the CB-IV adjoint into the adjoint of GRAPES-CUACE. After this development, uncertainties caused by different mechanisms will be carried out according to your valuable suggestion. The description of our current work is added in Section 5 Conclusions (lines 479-481).

*For comparing adjoint model results with Models-3/CMAQ simulations, the authors should use Source Apportionment (SA) tool attached to CMAQ rather than different sensitivity analysis CMAQ runs.*

Response: The Source Apportionment (SA) tool is a useful source sensitivity calculation tool. However, in our previous research (Zhai et al., 2016), through different sensitivity analysis CMAQ runs, we assessed emission sources control effects over the FLEXPART-determined sensitive source zones. Therefore, in this study, we compared the adjoint model results with different sensitivity analysis CMAQ runs in our previous study (Zhai et al., 2016).

***Minor comments***

*Abstract, Line 32 - 34 on Page 1: "This work also reflects ..." might be "The results imply that sensitive regional emissions reduction be more efficient than individual peak emission control for improving regional PM$_{2.5}$ air quality".*

Response: Thank you very much for your valuable comments. The manuscript has been revised according to your comments.

*Line 43 on Page 2: "Therefore, in concentration source sensitivity analysis problems," should be "Therefore, in concentration source sensitivity analysis, the adjoint method is more computationally efficient than others such as traditional finite difference method"*

Response: Revised.

*Line 57 on Page 2: "Research using approaches...have revealed that ..." should be "Researches using approaches ...have revealed that...".*

Response: Revised.

*Line 79 -80 on Page 3: "...during which time two PM$_{2.5}$ concentration peaks occur and are set as the objective functions." might be "during the episode two PM$_{2.5}$ concentration peaks ..."*

Response: Revised.

*Line 111 - 112 on Page 4: "Knowledge of the impacts of emission sources on pollutant concentrations can help enact effective air pollution control strategies." can be re-written into "Understanding of the impacts of emissions on*

*pollutant concentrations is helpful to develop effective air pollution control strategies.”*

Response: Revised.

*Line 133 - 134 on Page 5:” In Fig. S3, we can see that the adjoint sensitivity results are close to the finite difference results, ...”, How? I could not see it.*

Response: Below is Fig. S3. In Fig. S3, the blue line with circular symbols (the finite difference results) and the red line with triangle symbols (the adjoint sensitivity coefficients) are close, especially when the X-axis (PM$_{2.5}$ reduction ratios) are within 30%. To make presentation in the manuscript clearer, explanations are added in the revised manuscript at page 5, lines 133-134 in the parentheses.

[Figure]

**Figure S1. Comparisons of the adjoint sensitivity coefficients (red line with triangle symbols) and the finite difference results (blue line with circular symbols) for PM$_{2.5}$ primary emission reduction ratios at 5%, 10%, 20%, 30%, 50%, 70% and 90% over simulation domain for the Nov. 21 05:00, 2012 PM$_{2.5}$ peak.**

*Line 454 - 455 on Page 15: “Contributions to PM$_{2.5}$ concentration peaks from local Beijing and its surrounding provinces were compared.” may be replaced by “The peak PM$_{2.5}$ contributions from Beijing local emissions and neighbor provinces sources were well compared” if you like.*

Response: Following the referee's suggestion, it is changed in the revised manuscript.

**Reply to Referee # #3**

*The revised article by Zhai et al. includes several clarifications and updates. First, they clarify that their adjoint modeling does not include secondary aerosol formation. Sulfate and nitrate are instead treated as primarily emitted species. However, they do provide additional verification of the adjoint modeling results for primary species; these are linear and thus their interpretation of these sensitivities as being representative of large emissions perturbations is consistent with the forward model. These don't happen to be consistent with reality though, as sulfate and nitrate are not primary species. The potential impacts of these assumptions on their forward model evaluation is not considered.*

*Overall, I request that the authors remove sulfate and nitrate from their analysis and focus on their adjoint sensitivity analysis and source apportionment of primary species (BC, OC, dust, sea salt) until the point that their model is mature enough to include secondary species. Treating sulfate and nitrate as primary species, without any evaluation of the errors in this assumption, is not a good approach. Removing this problematic aspect would alleviate much of the*

*concerns from my review, and the other reviewer, regarding this paper.*

*Detailed comments in response to their revisions are provided below. Addressing these may involve another round of major revisions, to remove sulfate and nitrate from this study.*

Response: Thank you very much for your valuable comments.

In this study, the forward GRAPES-CUACE modelling system is an online coupled atmospheric chemistry modelling system, in which CUACE includes aerosol algorithm CAM, gaseous chemistry RADM II and the thermodynamic equilibrium ISORROPIA (described in Paragraph 1, Section 3.2). Therefore, in the forward simulation, both primarily emitted and secondarily formed sulfate and nitrate are considered. In the adjoint modelling, however, only primary sources of $PM_{2.5}$ were tracked, which are described in Paragraph 2, Section 3.2.

In the adjoint sensitivity analysis of this study, two hourly peak $PM_{2.5}$ concentrations were set as the objective functions, and the adjoint sensitivities imply primary emission sources impacts on the hourly peak $PM_{2.5}$ concentrations. As mentioned above, this study focuses on tracking primary emission sources of $PM_{2.5}$. As we are now coupling the CB-IV mechanism in the GRAPES-CUACE forward model, and embedding the CB-IV adjoint into the adjoint of GRAPES-CUACE, we will estimate sensitivities to both primary and precursor gaseous emission sources after this development.

This discussion is added in Section 5 Conclusions, lines 479-481.

***Specific comments:***

*- Because the authors do not include the adjoint of thermodynamic partitioning, they can not properly include secondary aerosols in their simulation. Their work-around for this is to treat sulfate and nitrate as primary species. This is very non-physical - sulfate and nitrate are secondary species, not emitted species, and any air quality model from the past two decades treats them as such. If they authors intend to introduce this major change in modeling, they need to quantify how much of an impact it makes on their forward model performance. This could be done, for example, by comparing complete forward model simulations (that include aerosol thermodynamics and secondary PM formation) with their primary-only version. Otherwise, their reference to earlier studies that have evaluated the performance of the forward model (e.g., Zhou et al., 2012; Want et al., 2015a, 2015b, Jiang et al., 2015) have no baring here.*

Response: Thank you for your comments. Although we didn't include thermodynamic partitioning in the adjoint simulation, thermodynamic partitioning and gaseous chemistry are included in the forward model, and secondary aerosol formation processes are considered in the forward simulation. That is, the forward model applied in this study is the same version as that in the work by Zhou et al. (2012), Wang et al. (2015a, 2015b) and Jiang et al. (2015). Tracking only primary emission sources of $PM_{2.5}$ (including sulfate and nitrate) with the adjoint model has no impact on the forward model performance.

Detailed descriptions of the forward model are in Paragraph 1, Section 3.2.

*- The statement that "adjoint sensitivity analysis are not significantly affected by overestimation of PM$_{2.5}$" is not correct. The accuracy of the adjoint model is directly related to accuracy of the version of the forward model for which the adjoint is created. This appears to be a version of the forward model that does not include secondary aerosol formation and instead treats sulfate and nitrate as primary emissions. This is not a forward model that is recommended, nor one that has been previously evaluated. While it is true that relative sensitivities from a linear model may be correct regardless of the absolute estimate of PM from the forward model, the authors would have to keep their study limited to the linear components of PM$_{2.5}$ (i.e., species that are actually primary in the atmosphere: BC, OC, dust and sea salt), for this statement to apply. It is also correct that the sensitivities from an adjoint model may match the same sensitivities in the forward model, as the authors show in Fig S3 — but this does not mean these sensitivities match real-world behavior.*

Response: As stated in the responses to previous comments, the forward model in this study includes the size-segregated multi-component aerosol algorithm CAM (Canadian Aerosol Module) as its aerosol module, the second generation of Regional Acid Deposition Model (RADM II) mechanism as its gaseous chemistry, and ISORROPIA to calculate the thermodynamic equilibrium between aerosols and their gas precursors. Therefore, the forward model includes secondary aerosol formation (Section 3.2, Paragraph 1), and is the version that has been previously evaluated by Zhou et al. (2012).

In this study, we only tracked the primary sources of PM$_{2.5}$, due to lack of secondary aerosol processes in the adjoint of GRAPES-CUACE modelling system. As it is validated in Fig. S3, PM$_{2.5}$ concentration and their primary sources have an approximately linear relationship. Under this circumstance, we assume that overestimation of PM$_{2.5}$ will not significantly affect the adjoint sensitivity analysis.

*- The authors' response to my question about the possible sources of uncertainty leading to overestimation of PM$_{2.5}$ concentrations by 200µg/m$^3$ was to provide a generic, obvious, list of factors that contribute to forward model performance: emissions, atmospheric processing, and meteorology. This provides zero insight into the nature of the uncertainty in their model. They do not specifically quantify or estimate how treatment of all PM$_{2.5}$ as primary may impact their results. They do not specifically quantify, estimate, or even mention the lack of heterogenous chemistry in even the complete forward GRAPES-CUACE model that has been implicated in several recent papers (see my previous review) as playing a key role in leading to peak PM$_{2.5}$ episodes.*

Response: In this study, the forward GRAPES-CUACE model is an online coupled atmospheric chemistry modelling system, in which CUACE includes aerosol algorithm CAM, gaseous chemistry RADM II and the thermodynamic equilibrium ISORROPIA (described in Paragraph 1, Section 3.2). The lack of heterogeneous chemical reactions (Wang et al., 2016; Cheng et al., 2016) in the forward GRAPES-CUACE model could be a factor causing the modeling uncertainties in this study, which is difficult to be quantified with the current version of forward GRAPES-CUACE model system. We are coupling the CB-IV mechanism in the GRAPES-CUACE forward model, and embedding the CB-IV

adjoint into the adjoint of GRAPES-CUACE. In future model development, we should try to include recently identified chemical reactions in chemical models to improve the simulation performance.

In the revised manuscript, we have added the above-mentioned discussions into Section 3.3, lines 221-223.

*- The fact that this modeling study only includes primary aerosols needs to be stated in the abstract and introduction, not just in the detailed description of the model (3.2).*

Response: This modeling study only considers primarily aerosol emission sources in its adjoint sensitivity analysis. In its forward simulation, however, both primary and secondary aerosols are considered. The adjoint sensitivities of $PM_{2.5}$ to only primarily emitted emission sources are added in the abstract (P1, line 17; P1, line 24; P1, line 25) and introduction (P3, line 78).

*- Section 3.2 needs to explicitly state that the authors do not include an adjoint of aerosol thermodynamic partitioning — this still is not explicitly stated.*

Response: It is now added in Paragraph 2, Section 3.2 (lines 163-165) following the referee's suggestion:

As the adjoint of the gaseous chemistry (RADM II) and the adjoint of the thermodynamic equilibrium (ISORROPIA) processes are not included in the GRAPES-CUACE aerosol adjoint model, the GRAPES–CUACE aerosol adjoint model is capable of coupling major aerosol processes contained in CAM (described in the above paragraph) into its simulations of the sensitivities of the objective function to primary aerosol sources.

*- The phrase "rigorous mathematical verification" redundant and vague. Any type of mathematical verification is rigorous by definition, yet this statement still fails to convey the essence of what has been verified. The test they have performed is referred to in the adjoint modeling community as the Lagrange condition; that would be a more informative way to describe this evaluation.*

Response: We have clarified this statement in the revised manuscript (lines 183-185) as followings:

After the tangent linear model (TLM) and the adjoint model are built (the adjoint model is a concomitant of the TLM), they are divided into smaller sections and tested separately before the assembled TLM and the adjoint model are confirmed valid. Details of the TLM and adjoint verification can be found in An et al. (2007).

*- The additional finite difference tests are appreciated, and show that the adjoint sensitivities are rather consistent. The consistency of the sensitivities across different perturbation magnitudes is a consequence of the linearity of primary $PM_{2.5}$. It does not match the known non-linear response of actual $PM_{2.5}$ concentrations in China to changes in $SO_2$ and $NOx$ emissions (e.g., Wang et al., ES&T, 45, 9293-9300, 2011).*

Response: As the secondary aerosol formation processes were not considered in the current GRAPES-CUACE aerosol adjoint model, only primary emission sources are tracked in its adjoint sensitivity analysis. We are developing the adjoint of gaseous chemistry in the GRAPES-CUACE adjoint model. After this development, the non-linear response of actual $PM_{2.5}$ concentrations to their precursor gaseous emissions could be explored.

[revised manuscript text omitted]

---

## Author Response (AR4)

Dear co-editor and referees,

Thank you very much for your valuable comments. Accordingly, we have completely revised the manuscript. Please see the revisions in red marked or highlighted texts in the revised manuscript. Responses to your comments are as follows:

*Co-Editor Decision: Reconsider after major revisions (19 Dec. 2017) by Renyi Zhang*
*Comments to the Author:*
*Please carefully address the recommendations by this reviewer. Specifically, the reviewer still considered that you were vague on how your source attribution calculations with the adjoint treat sulfate and nitrate as primary species. The reviewer urged you to focus on just the truly primary species or to include a more complete description and verification of how they treated sulfate and nitrate as primary species in the adjoint. Also, I ask you to provide a comprehensive assessment of the uncertainties in your work, including those with emissions, meteorology, and chemistry. In addition, I agree with this reviewer that your manuscript requires careful proofreading to improve its readability.*

Response: Thank you very much for your comments. In this study, we are in fact focusing on just the primary sources of $PM_{2.5}$. Primary sulfate and primary nitrate sources are allocated from the primary $PM_{2.5}$ sources in the emission inventories constructed by Cao et al. (2011). More details regarding information of primary $PM_{2.5}$ emissions and their allocation methods are described in answers to the referee's comment. To avoid misunderstanding, we have revised the improper expression of 'BC, OC, primary sulfate, primary nitrate and other fine particulate matters' throughout the manuscript to 'primary $PM_{2.5}$', consistent with the introduction of the emission inventories.

Performance of $PM_{2.5}$ concentration simulation is evaluated in Section 3.3. In addition, we added model evaluation of the 2 m temperature (T2m) and the 10 m wind speed (WS10m) in Section 3.3 as the following description in the revised manuscript:

The reliability of the GRAPES-CUACE modeling system is evaluated in terms of both meteorological and chemical simulations. Figure 5 shows the hourly variations of the observed and simulated 2 m temperature (T2m) and 10 m wind speed (WS10m), and Table 1 lists the corresponding statistical parameters. The correlation coefficients (Rs) between the observed and simulated hourly T2m are 0.77, 0.75 and 0.74, passing the 99% confidence level with root mean square error (RMSE) values of 1.5, 1.6 and 1.7 °C, respectively at observatory sites Nanjiao (NJ), Haidian (HD) and Shangdianzi (SDZ). Mean Bias (MB) values for the T2m demonstrate a slight underestimate in NJ (-0.1 °C) and HD (-0.3 °C), and overestimate in SDZ (0.8 °C). The variations of the WS10m are generally captured by the model with Rs

of 0.70, 0.73 and 0.46, and with RMSEs of 1.4, 1.5 and 1.8 m s$^{-1}$ at NJ, HD and SDZ stations respectively (passed the 99% confidence level). Overall, the GRAPES-CUACE could reasonably reproduce the observed meteorology.

[Figure]

Figure 5 The temporal variations of observed and simulated hourly 2 m temperature (a-c) and 10 m wind speed (e-f) at Nanjiao, Haidian and Shandianzi stations

Table 1 Statistics between observed and simulated meteorology.

| | Nanjiao | | | | | Haidian | | | | | Shangdianzi | | | | |
|---|---|---|---|---|---|---|---|---|---|---|---|---|---|---|---|
| | Obs. | Mod. | MB | R | RMSE | Obs. | Mod. | MB | R | RMSE | Obs. | Mod. | MB | R | RMSE |
| T (℃) | 4.2 | 4.1 | -0.1 | 0.77 | 1.5 | 3.6 | 3.3 | -0.3 | 0.75 | 1.6 | 1.0 | 1.8 | 0.8 | 0.74 | 1.7 |
| WS (m s$^{-1}$) | 1.9 | 2.4 | 0.5 | 0.70 | 1.4 | 1.5 | 2.4 | 0.9 | 0.73 | 1.5 | 1.9 | 2.6 | 0.6 | 0.46 | 1.8 |

*Suggestions for revision or reasons for rejection (will be published if the paper is accepted for final publication)*

*The authors have made it more clear that the forward model includes secondary aerosol formation.*

*As the primary results of this paper though are the source attribution results from the adjoint model, it still suffices to say they're not really treating secondary PM$_{2.5}$ in their present study — even if yes it is included in the forward model. So I appreciate that they have qualified their results as pertaining to "primary" species throughout the article.*

*However, I was expecting more. The approximation necessary for constructing the adjoint model using primary tracers for sulfate and nitrate are not explicitly stated, and some details and evaluation are missing. For example, how do the authors construct an emission inventory of aerosol nitrate? Do they assume a particular fraction of NOx becomes nitrate? Same question for sulfate and SO$_2$? The authors need to be very clear about these sorts of details in order for the interpretation of their sensitivity results for these species to be meaningful; or they should leave out these species and focus on the primary species. They should still demonstrate, with their forward model alone, that sulfate and nitrate can be treated as primary species for these case studies.*

Response: Thank you very much for your comments. We only considered the primary PM$_{2.5}$ in our present adjoint simulation, and the adjoint of the gas-to-particle processes is under construction now. In this study, detailed high-resolution emission inventories of primary PM$_{2.5}$ and pollutant gases for China in 2007 were used (Cao et al., 2011).

In order to establish the emission inventories applied for the GRAPES-CUACE model, referring to the study by Fu et al. (2013), the primary PM$_{2.5}$ were further allocated to OC, BC, primary nitrate, primary sulfate and other fine particulate matters. Therefore, the sources of primary sulfate and nitrate in this work are in fact allocated from the primary PM$_{2.5}$ in the emission inventories created by Cao et al. (2011). For example, OC, BC, primary nitrate, primary sulfate and other fine particulate matters accounts for 22.5%, 11.0%, 0.7%, 8.3% and 57.6% of that of primary PM$_{2.5}$ sources. The adjoint sensitivities analyzed in this study refer to the gradients of the objective PM$_{2.5}$ concentrations to the primary PM$_{2.5}$ sources. To correct the presentation, we revised the expression of the sources considered in this study to 'primary PM$_{2.5}$ sources' in the manuscript.

To clarify the the presentation, we have made the following changes in the manuscript:

1) We rewrite the sentence in Paragraph 2, Section 3.2 as: Hence S$_n$ defined in section 3.1 refers to primary PM$_{2.5}$.

2) Rewrite the sentence in Paragraph 2, Section 3.3 as: Anthropogenic emissions include primary PM$_{2.5}$ and pollutant gases.

3) Change 'primary particulate sources' to 'primary PM$_{2.5}$ sources' in Paragraph 2, Section 3.1 and in the last Paragraph of Section 4.5.

4) Change 'primary particle sources' to 'primary PM$_{2.5}$ sources' in Paragraph 3, Section 3.3.

Reference:

Fu X, Wang S, Zhao B, et al. Emission inventory of primary pollutants and chemical speciation in 2010 for the Yangtze River Delta region, China. Atmospheric Environment, 2013, 70(70):39-50.

*It's not clear to me whether or not the full forward model including ISORROPIA etc was used for the finite difference perturbation calculations included in figure S3 — the description of this critical verification test is quite limited — we aren't told which species emissions are being reduced — all of them? It's just very hard to imagine that in the full forward model a 90% change in NOx emissions would elicit a linear PM$_{2.5}$ response. If that is true in their model, then they need to explain why their model differs from other previous studies examining the non-linear response of PM$_{2.5}$ to NOx emissions changes.*

Response: The full forward model including ISORROPIA was used for the finite difference perturbation

calculations in Fig. S3, but only primary $PM_{2.5}$ were reduced in the emission source sensitivity tests. We reduced the primary $PM_{2.5}$ sources by 5%, 10%, 20%, 30%, 50%, 70% and 90% over simulation domain, and found that the peak $PM_{2.5}$ concentrations decreased linearly (Fig. S3, blue line with circular symbols), indicating a linear relationship between $PM_{2.5}$ concentrations and primary $PM_{2.5}$ sources.

We have added a paragraph in the supplement to specifically describe this verification:

Figure S3 illustrates comparisons of the adjoint sensitivity coefficients (red line with triangle symbols) and the finite difference results (blue line with circular symbols). For the finite difference tests, the primary $PM_{2.5}$ sources are reduced by 5%, 10%, 20%, 30%, 50%, 70% and 90% accordingly, and the Y-axis indicates the decreased $PM_{2.5}$ concentrations due to corresponding emission reductions. For the adjoint sensitivity results, Y-axis indicates the time cumulated adjoint sensitivity results summed over the entire simulation domain multiply by emission reduction ratios on the corresponding X-axis. According to the adjoint sensitivity coefficients defined in Section 3.1, Y-axis for the adjoint sensitivity results indicates the changes in $PM_{2.5}$ concentration due to certain proportion (on the X-axis) of perturbations in primary $PM_{2.5}$ emissions.

*- The first reviewer's comment about the linearity of the adjoints only being correct for calm systems isn't correct. That reviewer is mistakenly thinking of the adjoint of a weather model; in contrast here the active variables are only tracers. The transport of tracers is always linear, regardless of calm or dynamic systems. The non-linearities of concern are those related to the chemical environment.*

Response: Thank you very much for pointing this out. We have deleted the sentence added in Section 5.

*- 165: A critical correction, the "GRAPES-CAUCE aerosol adjoint model is NOT capable of coupling major aerosol processes in CAM"…*

Response: We have changed this sentence to: "the GRAPES-CUACE aerosol adjoint model is capable of simulating sensitivities of the objective function to primary $PM_{2.5}$ sources".

*- the manuscript requires grammatical proofreading throughout*

Response: The manuscript has been edited by the Enago (http://www.enago.com) English editing services.

[revised manuscript text omitted]

**Figure 9**. Hourly variations of surface PM$_{2.5}$ concentrations in Beijing and sensitivity coefficients of surface PM$_{2.5}$ concentration peaks in Beijing to local and surrounding primary PM$_{2.5}$ sources. The left and right panels correspond to PM$_{2.5}$ concentration peaks at 05:00 LT and at 23:00 LT on the 21st of November 2012, respectively. (a–b) Hourly variations of Beijing PM$_{2.5}$ concentrations (black solid dot-line) and hourly instantaneous sensitivity coefficients to local (red closed squares) and surrounding (red open squares) emission sources. (c–d) The time-integrated sensitivity coefficients to local (red closed squares) and surrounding (red open squares) emission sources.

[Figure]

**Figure 10.** Sensitivity coefficients of surface PM2.5 concentration peaks in Beijing to primary emission sources from local Beijing and each of the surrounding provinces. The left and right panels correspond to PM2.5 concentration peaks at 05:00 LT and at 23:00 LT on November 21, 2012, respectively. (a–b) Hourly instantaneous sensitivity coefficients to emission sources from local Beijing, Hebei Province, Tianjin City, Shanxi Province, and Shandong Province. (c–d) The time-integrated sensitivity coefficients to local and surrounding provincial emission sources. (e–f) The contribution ratios of emission sources from each surrounding province to PM2.5 concentration peaks.

[Figure]

**Figure 11.** The 24 (a), 48 (b), and 72 h (c) integrated sensitivity coefficients of surface PM2.5 concentrations to primary emission sources in Beijing on November 21, 2012.

[Figure]

| Regions | Number of grid cells | Sensitive area ratios (%) |
|---|---|---|
| HuaB-sens | 18 | 10.2 |
| HuaB | 176 | |
| BJ-sens | 6 | 60.0 |
| BJ | 10 | |
| Emis-intense | 18 | 10.2 |

**Figure 12.** Domain definition of Huabei (HuaB, in red dot-dashed frame), Beijing (BJ, in black solid frame), sensitive Beijing (BJ-sens, red shaded), sensitive Huabei (HuaB-sens, both red and blue shaded), and emission intensive (Emis-intense, in pink solid frame) regions.

Notes: HuaB-sens area ratio = HuaB-sens floor space/HuaB floor space × 100%;

BJ-sens area ratio = BJ-sens floor space/BJ floor space × 100%;

Emis-intense area ratio = Emis-intense floor space/HuaB floor space × 100%.

---

## Author Response (AR5)

Dear Co-Editor and referee,

Thank you very much for your valuable comments. Please find the revisions in red marked or highlighted texts in the revised manuscript. Responses to your comments are as follows:

*The manuscript presents an interesting modeling work to identify the primary emission sources in Beijing using an adjoint model. As an added reviewer, I think the revised manuscript got substantially improved following the suggestions in the previous reviews, and it can be accepted by ACP. Some additional minor revisions are needed to enhance the readability of the paper.*

Response: Revisions are made according to your comments listed below.

*The adjoint theory and adjoint model are mentioned in the very beginning of the paper. However, their backgrounds are not well introduced. Many associated technical terms, such as "receptor-oriented", "objective function", "unequilibrated variables" are hard to understand for most readers. I suggest the authors use more layman's language to discuss the importance of an adjoint model.*

Response: Thank you very much for pointing this out. We have revised the beginning of Section 1 (the Introduction Section) with the technical terms "receptor-oriented" and "source-oriented" well explained. We have also revised the term "objective function" to "cost function", a much more widely used term in previously published articles, throughout the whole manuscript. We are not using the term "unequilibrated variables" now, and it is changed to the description in Section 3.2 as "the model state variables (concentrations)". The term "unequilibrated variables saving" in Figure 4 (operational processes of the GRAPES-CUACE aerosol adjoint) is also revised to "state variables saving" accordingly. Hope these revisions can make the manuscript more readable.

*Also, Fig. S1 and S2 are illustrative for the method used in this study, so I recommend moving them to the main body of the paper.*

Response: Figures S1 and S2 are respectively moved to Figures 2 and 3 in the main body of the paper.

*Move Figure 4 to Figure 1, as it is about fundamental geospatial information.*

Response: Following the suggestion, Figure 4 is moved to Figure 1.

*Figure 5 caption, please specify what observations are used for 2-m temp. and 10-m wind.*

Response: The observations of 2-m temp and 10-m wind are collected from the weather monitoring

network of Beijing Meteorology Bureau. We have added a sentence in the manuscript in Section 3.3 to illuminate this. In Figure 5 (now Figure 7) caption, we clarified that the observed WS10m are 10-min averaged wind speed. In addition, the Y-axis annotations are changed to "T2m" and "WS10m" for clarity.

*Figure 6, why there is no observation for PM$_{2.5}$ in Beijing?*

Response: Figure 6 is now moved to Figure 8. Figure 8d shows the simulated average PM$_{2.5}$ concentration over Beijing Municipality, and it is used to illustrate the hourly variation of PM$_{2.5}$ concentration over the entire Beijing.

*Other revisions:*

1. Two sentences are added in the conclusion part:

[revised manuscript text omitted]